# Sensitivity of biomass burning emissions estimates to land surface information

Makoto Saito[1*], Tomohiro Shiraishi[1], Ryuichi Hirata[1], Yosuke Niwa[1], Kazuyuki Saito[2],
Martin Steinbacher[3], Doug Worthy[4], and Tsuneo Matsunaga[1]

[1]National Institute for Environmental Studies, 16-2 Onogawa, Tsukuba, Japan
[2]Japan Meteorological Agency, 1-3-4 Ote-machi, Chiyoda-ku, Tokyo, Japan
[3]Swiss Federal Laboratories for Materials Science and Technology, Ueberlandstrasse 129, CH-8600 Duebendorf, Switzerland
[4]Environment and Climate Change Canada, 4905 Dufferin Street, Toronto, Ontario Canada

**Correspondence:** M. Saito (saito.makoto@nies.go.jp)

**Abstract.** Emissions from biomass burning (BB) are a key source of atmospheric tracer gases that affect the atmospheric carbon cycle. We developed four sets of global BB emissions estimates (named GlcGlob, GlcGeoc, McdGlob, and McdGeoc) using a bottom-up approach and by combining the remote sensing products related to fire distribution with two aboveground biomass (AGB) and two land cover classification (LCC) distributions. The sensitivity of the estimates of BB emissions to the AGB and LCC data was evaluated using the carbon monoxide (CO) emissions associated with each BB estimate. Using the AGB and/or LCC data led to substantially different spatial estimates of CO emissions, with a large (factor of approximately three) spread of estimates for the mean annual CO emissions; $526 \pm 53$, $219 \pm 35$, $624 \pm 57$, and $293 \pm 44$ Tg CO yr$^{-1}$, for GlcGlob, GlcGeoc, McdGlob, and McdGeoc, respectively, and $415 \pm 47$ Tg CO yr$^{-1}$ for their ensemble average (EsmAve). We simulated atmospheric CO variability at approximately 2.5° grid using an atmospheric tracer transport model and the BB emissions estimates and compared it with ground-based and satellite observations. At ground-based observation sites during fire seasons, the impact of intermittent fire events was poorly defined in our simulations due to the coarse resolution, which obscured temporal and spatial variability in the simulated atmospheric CO concentration. However, when compared at the regional and global scales, the distribution of atmospheric CO concentrations in the simulations show substantial differences among the estimates of BB emissions. These results indicate that the estimates of BB emissions are highly sensitive to the AGB and LCC data.

## 1 Introduction

The majority of biomass burning (BB) is related to human activities, with only a small fraction caused by natural processes such as lightning (Seiler and Crutzen, 1980; Balch et al., 2017). Various agricultural and economic processes involve BB; e.g., clearing of forest and brush land for agricultural use, or controlling fuel accumulation in forests (Andreae, 1991). Such intensive activities have significant implications for changes in regional land cover from fire-resistant to fire-prone systems (Turetsky et al., 2015). Even in savanna where fire-adapted trees are dominant, frequent fires and/or an abrupt increase in fire intensity can result in ecosystem degradation with a subsequent reduction in woody biomass (Saito et al., 2014). Furthermore,

BB is a significant source of trace gases and aerosol particles in the atmosphere (e.g., Bougiatioti et al., 2014; Pan et al., 2020). Water vapor and carbon dioxide ($CO_2$) are the primary products of the burning of organic materials. In addition, in incom-plete combustion, various other compounds such as carbon monoxide (CO), methane ($CH_4$), nitrogen oxides, and ammonia are emitted from the fires (Andreae, 1991). Recent studies have shown that climate change associated with rising anthro-pogenic emissions of greenhouse gases might lead to an increase in fire frequency over some regions (e.g., boreal regions), and emissions of greenhouse gases from terrestrial biosphere due to enhancement of BB (Dutta et al., 2016; Hart et al., 2019).

The quantification of BB emissions is crucial to our understanding of the role of BB with respect to the global carbon cycle and its interaction with climate change. At present, several global BB emission inventories are available thanks to advances in satellite observation technology. The satellite remote sensing products for fires, such as fire radiative power, active fires, and burned area, have been used to develop inventories of BB emissions (e.g., Wiedinmyer et al., 2011; Kaiser et al., 2012; van der Werf et al., 2017). These inventories provide BB emissions covering a variety of temporal and spatial resolutions and have facilitated improvements in our understanding of greenhouse gas emissions from BB. Research efforts, including the continuous improvement of land surface information, are important to the study of the global carbon cycle, as they have reduced estimation uncertainties. However, estimates of annual greenhouse gas and aerosol particle emissions based on these inventories still show significant differences of $1.5\times$ and $3.8\times$ for $CO_2$ and organic carbon, respectively, as well as variations in the spatiotemporal patterns of the emissions (Shi et al., 2015; Pan et al., 2020). These divergences mean that our ability to represent BB processes at the global scale remains limited, and a basic research question regarding which of the available estimates better represents BB emissions over regional and global scales remains to be answered. Consequently, there is further need for information related to the development of better estimates of BB emissions.

Our analysis of BB emissions inventories used a bottom-up approach and incorporated land surface information regarding vegetated biomass density and land cover classifications (LCC), information on fire events, and several parameters related to burning efficiency. Although land use mapping has improved over recent decades (e.g., Popescu et al., 2011; Rodriguez-Galiano et al., 2012), differences of between 10% and 20% remain in estimates of regional carbon stocks based on different land surface maps (Mitchard et al., 2013), which can cause divergence in estimates of BB emissions. In this study, we aim to evaluate the sensitiv-ity of estimates of BB emissions to land surface information and compare our findings with the newly proposed BB emissions estimates. A detailed description of our BB emissions estimates, as well as a broad summary of the comparison of our estimates with four published reference BB datasets covering $CO_2$ emissions, can be found in Shiraishi et al. (2021). This study takes an additional step towards evaluation of BB estimates in terms of atmospheric CO variability, which can be used as a tracer to investigate the transport of BB emissions (e.g., Chen et al., 2009; Mu et al., 2011), using independent reference data from ground-based and satellite observations of atmospheric CO concentrations.

## 2   Methods

### 2.1   Biomass burning estimates

This study expresses CO emissions from BB ($E$; g CO month$^{-1}$) at a grid ($i$) at a resolution of 500 m with the LCC ($j$) in each month ($k$), using the burned area method (e.g., Michel et al., 2005; Mieville et al., 2010):

$$E_{i,k} = BA_{i,k} \cdot F_{i,k} \cdot EF_j, \tag{1}$$

$$F_{i,k} = BE_j \cdot \sum_{l=m+1}^{n} \left(AGB_i(1-BE_j)^{l-1}\right) \tag{2}$$

where $BA, F, EF, BE, \mathrm{and} AGB$ are the burned area (m$^2$), the flammable fuel (kg m$^{-2}$), the emission factor (g CO kg$^{-1}$), the
burning efficiency (which ranges from 0 to 1), and the above-ground biomass (kg m$^{-2}$), respectively. The values of $EF$ and $BE$ are defined by LCC (Tables A1 and A2). The parameters $m$ and $n$ are the cumulative number of fire occurrences during the previous ($k-1$) and current month $k$, respectively. Equation (2) represents decreases in $F$ owing to reductions of $AGB$ by frequent fires in a year. The biomass density; i.e., flammable fuel, decreases with increasing fire occurrence $l$. Note that the largest values of $E$ and $F$ occur during the first fire event in a year, as shown in Eqs (1) and (2), then $E$ and $F$ decline as more
fire events occur. $AGB$ is reset to its original magnitude; i.e., before the fires, at the beginning of each year.

To determine the sensitivity of the BB emissions estimates to the land surface information used, we calculated $E$ based on four scenarios that combined two types of LCC and two types of AGB data. The LCC maps derived from the Global Land Cover 2000 project (GLC2000) (Bartholomé and Belward, 2005) and the Moderate Resolution Imaging Spectroradiometer (MODIS) Land Cover Type (MCD12Q1) Version 6 (Sulla-Menashe et al., 2019) data products were used to classify the land cover
types in each grid. The GLC2000 provides a global LCC map with 22 land cover types (Table A1) based on daily data from the VEGETATION sensor onboard the Satellite Probatoire de l'Observation de la Terre (SPOT-4) satellite. It covers 14 months from 1 November 1999 to 31 December 2000 with a 1-km spatial resolution. We also used the MCD12Q1 International Geosphere-Biosphere Programme (IGBP) legend as another LCC map. This product provides a global LCC map with 17 land cover types (Table A2) with a spatial resolution of 500 m and yearly temporal resolution after 2001. The AGB maps were obtained from
two datasets: the GEOCARBON global forest biomass map (Avitabile et al., 2016) and the Globbiomass AGB map (Santoro, 2018). The GEOCARBON map is a combined AGB map based on two pan-tropical datasets published by Saatchi et al. (2011) and Baccini et al. (2012) with reference field data and biomass maps, and provides the global AGB map at a 1-km spatial resolution. The Globbiomass map is an AGB product based on satellite observations from the radar backscattered intensity recorded by the Phase Array-type band Synthetic Aperture Radar (PALSAR) instrument, which is onboard the Advanced Land
Observing Satellite (ALOS), and the Advanced Synthetic Aperture Radar (ASAR) instrument operating at C-band, which is onboard the Environmental Satellite (Envisat) and uses LiDAR-based metrics and surface reflectances. This AGB product is produced by the European Space Agency (ESA) with a 25-m spatial resolution. These LCC and AGB maps were used in Eqs (1) and (2) by aggregating or disaggregating them to a spatial resolution of 500 m.

The BA was obtained from the MODIS Thermal Anomalies and Fire Daily (MOD14A1) Version 6 dataset (Giglio et al., 2016). MOD14A1 provides daily fire mask compositions at a 1-km resolution, and we used the low-, nominal-, and high-confidence fire classes (FireMask = 7, 8, and 9, respectively) to detect BA. This study disaggregated the MOD14A1 product into a 500 m resolution, and assigned $BA_{i,k} = 250,000$ m$^2$ for any grid showing a fire flag. Fire occurrences over a grid $i$ in a month $k$ were not involved in BA in this study, but they vary $E$ with changes of $F$ in Eq. (2). The fire occurrences were determined by counting the number of discontinuities of fires in a month. This means that, if the FireMask shows flags for fires (FireMask = 7,8,9) continuously over a month, the fire occurrence was set to $l = 1$.

The EF for CO was derived from the study of van der Werf et al. (2017). They compiled an EF dataset for six biomes based on the studies of Andreae and Merlet (2001) and Akagi et al. (2011). For this study, we reallocated the EF to the 22 land cover types used in GLC2000 (Table A1) and the 17 LCC types used in MCD12Q1 (Table A2). In this process, we classified the globe into 14 regions (after Giglio et al. (2006) and van der Werf et al. (2017); Fig. A1), then the EF from the 6 biomes was adapted to the corresponding LCC types based on the location of the objective grid in a region. The BE was derived from the study by Mieville et al. (2010). As their LCC conformed to GLC2000 (Table A1), the BE values were assigned to 17 LCC types on the MCD12Q1 map (Table A2).

We hereafter refer to BB emissions estimates based on GLC2000 using Globbiomass and GEOCARBON as GlcGlob and GlcGeoc, respectively, and those based on MCD12Q1 using Globbiomass and GEOCARBON as McdGlob and McdGeoc, respectively (Table 1). Further, to obtain a single estimate from these four estimates, an ensemble average (EsmAve) of the four BB emissions estimates is used. This value approximates the center of the probability distribution of BB emissions estimates derived from two AGB and two LCC datasets. We used $E$ data that were aggregated onto a grid with a resolution of about 0.837 degree in the following analysis.

## 2.2 Atmospheric tracer transport model

We used the Non-hydrostatic ICosahedral Atmospheric Model (NICAM)-based Transport Model (NICAM-TM; Niwa et al., 2011) to simulate atmospheric CO concentrations in this study. NICAM has a unique characteristic in its dynamical core; i.e., a non-hydrostatic system in the flux form that guarantees the conservation of tracer mass (Satoh, 2002). NICAM implements this non-hydrostatic scheme using an icosahedral grid configuration.

NICAM-TM includes a module for the reaction processes among hydroxyl radical (OH) and CO, and the oxidation of CH$_4$ with OH, which yields CO, to simulate atmospheric CO variability (Niwa et al., 2020). We used atmospheric OH field data from the TransCom-CH$_4$ project (Patra et al., 2011). The atmospheric CH$_4$ concentration in the simulation was fixed at 1800 ppb, which is the maximum value of whole atmospheric CH$_4$ concentration derived from the Greenhouse gases Observing SATellite observations (Yokota et al., 2009) for 2009 and 2015. In this study, we used a globally uniform grid system with a horizontal resolution of about 220 km and 40 vertical layers. Horizontal winds in NICAM-TM are nudged using the Japanese 55-year Reanalysis (Kobayashi et al., 2015) to simulate substantial atmospheric transport. We used the NICAM-TM version described by Niwa et al. (2017) for the transport of CO.

Fossil fuel, biogenic, and biomass burning CO-emission-inventories were used as the CO emission sources at the Earth's surface. The fossil fuel CO emissions were derived from the Emissions Database for Global Atmospheric Research (EDGAR v4.3.2; Janssens-Maenhout et al., 2019) with an annual resolution. Biogenic CO emissions from vegetation were derived from a process-based model, the Vegetation Integrative SImulator for Trace gases (VISIT; Ito, 2019). The biogenic CO emissions in VISIT are simulated as a part of processes associated with biogenic volatile organic compound emissions and have a monthly resolution. For CO emissions from BB, the abovementioned four scenarios are based on the various combinations of the LCC and AGB maps.

## 2.3 Observational data

Ground-based observations of atmospheric CO concentrations were downloaded from the World Data Centre for Greenhouse Gases (WDCGG) for 2009–2015. From the 28 WDCGG stations with hourly observation data, we selected those showing abrupt increases in atmospheric CO concentrations during dry seasons. Consequently, two ground-based sites: Bukit Kototabang, Indonesia (BKT, 0.20°S, 100.32°E; Zellweger et al., 2019) and East Trout Lake, Canada (ETL, 54.35°N, 104.99°W; Kim, 2016) were selected to evaluate the estimates of BB emissions from local fire events. Minamitorishima, Japan (MNM, 24.29°N, 153.98°E; Watanabe et al., 2000) was also used as spatially representative background information. The MNM site is situated on a remote coral island in the western North Pacific where the influence of local fire events is usually not significant, because there is no other island for over 1,000 km in any direction, whereas the data from the BKT and ETL sites may be influenced by wildfires.

We also used the column-averaged dry-air concentration of CO (XCO; ppb) recorded by the Measurements of Pollution in Troposphere (MOPITT) (Deeter et al., 2003) instrument on NASA's Earth Observing System Terra platform. The monthly mean XCO distribution was calculated by dividing the retrieved CO total column data (mol cm$^{-2}$) by the dry air column data (mol cm$^{-2}$) in the MOPITT Version 9 Level 3 multispectral (thermal infrared and near-infrared) products (MOP03JM-L3V95.6.3; Deeter et al., 2014, 2021) for the period 2013–2015. The XCO was calculated separately for daytime and nighttime MOPPIT observations, and the average of these data were used in this study.

## 2.4 Application of MOPITT averaging kernels to simulations

To compare our simulation results with MOPITT XCO, MOPITT averaging kernels and a priori information used in the MOPITT retrievals were incorporated as follows into the simulated CO total column $c_s$ with units of mol cm$^{-2}$ (Deeter, 2002):

$$c_s = c_a + \mathbf{a}(\mathbf{x_s} - \mathbf{x_a}),$$ (3)

where $c_a$, $\mathbf{a}$, $\mathbf{x_s}$, and $\mathbf{x_a}$ are a priori CO total column data, the CO total column averaging kernel, and simulated and a priori CO profiles, respectively. The CO profiles $\mathbf{x}$ are described using the common logarithm of CO volume mixing ratios $\mathbf{v}$, $\log_{10} \mathbf{v}$, at particular levels in the profile. Both $\mathbf{a}$ and $\mathbf{x}$ are vectors, and $c$ is a scalar. The a priori CO total column $c_a$ is calculated by multiplying the total column operator $\mathbf{t}$ and $\mathbf{x_a}$:

$$c_a = \mathbf{t^T x_a},$$ (4)

where the superscript $\mathbf{T}$ denotes the transpose operator. The total column operator $\mathbf{t}$ defines the layer width between the particular pressure levels $\mathbf{p}$,

$$\mathbf{t} = k\mathbf{\Delta p}, \tag{5}$$

where $k$ is the conversion factor between $\mathbf{v}$ and the column amount ($k = 2.120 \times 10^{13}$ mol cm$^{-2}$ hPa$^{-1}$ ppb$^{-1}$). Surface pressure $p_{sfc}$ and nine fixed pressure levels of 900, 800, 700, 600, 500, 400, 300, 200, and 100 hPa are used as the MOPITT retrieval levels. For $p_{sfc}$ higher than the bottom layer of 900 hPa, a 10-level vertical layering is used in Eqs (3)–(5), whereas the number of layers decreases as $p_{sfc}$ decreases below 900 hPa. The total column averaging kernel is determined as follows using $\mathbf{t}$ and the averaging kernel matrix $\mathbf{A}$:

$$\mathbf{a} = \mathbf{t}^{\mathbf{T}}\mathbf{A}. \tag{6}$$

We used the values of $\mathbf{x_a}$, $p_{sfc}$, and $\mathbf{A}$ derived from the MOPITT level 3 product, while $\mathbf{x_s}$ is the simulated CO concentrations using NICAM-TM with the BB emissions estimates. As all parameters in the MOPITT products are provided with a spatial resolution of 1° grid, the simulated monthly CO concentrations with a 2.5° spatial resolution were interpolated to 1° grids. Then $c_s$ values for daytime and nighttime were calculated using the corresponding MOPITT parameters, and converted to XCO using the corresponding MOPITT dry air column data. The average values of daytime and nighttime XCO were used as the simulated XCO.

## 2.5 Reference inventories

To quantify uncertainties in the spatial distributions of BB emissions, we compared our BB emissions estimates to two widely used inventories: the Global Fire Emissions Database (GFED4.1s; van der Werf et al., 2017) and the Global Fire Assimilation System (GFASv1.2; Giuseppe et al., 2018). GFED4.1s estimates 3-hourly BB emissions at 0.25° grid using the burned area method, but the emissions from small fires are also combined using information about thermal anomalies and surface reflectance derived from MODIS products. Fuel components in GFED4.1s are based on a terrestrial biosphere model, called the Carnegie–Ames–Stanford Approach (CASA; Potter et al., 1993). GFASv1.2 estimates daily BB emissions at 0.1° grid based on fire radiative power from MODIS products. This estimate models fire persistence by considering weather conditions using the fire weather index modeling (Van Wagner et al., 1987).

## 2.6 Modified index of agreement and standardized anomaly

We used the modified index of agreement (MIA) (Willmott et al., 1985) to compare the observed and simulated atmospheric CO concentrations, as follows:

$$\text{MIA} = 1.0 - \frac{\sum_{i=1}^{N}|x_i - y_i|}{\sum_{i=1}^{N}(|y_i - \overline{x}| + |x_i - \overline{x}|)} \tag{7}$$

where $x$ and $y$ are the observed and simulated CO concentrations (ppm) and $\overline{x}$ is the sample mean of $x$. The MIA calculates normalized value from 0.0 to 1.0 with higher values indicating better agreement between the observations and the model

simulations. Correlation coefficient indicates higher value for agreement of phase variations in the variability, whereas the
MIA does for both agreements of phase and amplitude gain variations in the variability.

The observational time series from the BKT and ETL sites were used to classify the 'no fire' or 'fire' months based on the standardized anomaly $z$:

$$z_i = (x_i - \overline{x})/\sigma_x \tag{8}$$

where $x$ is the observed daily CO concentration (ppb) and $\sigma_x$ is the corresponding sample standard deviation. In this study, fire
months were empirically identified as having observed CO concentrations corresponding to $z_i \geq 1.5$.

## 3  Results

### 3.1  Comparisons of emissions estimates

AGB is a source of flammable fuels for BB in our estimate. A comparison of the two AGB datasets (i.e., GEOCARBON and Globbiomass) for 2009 and the cumulative probabilities within the range of biomass availability of $0 < \text{AGB} \leq 40 \text{ kg m}^{-2}$
are shown in Fig. 1. The distribution of AGB differs between the two products (Fig. 1a), but there is a relationship between them with a correlation of $r = 0.93$. AGB is most often less than 5 kg m$^{-2}$ in both AGB products. AGB availability of $\leq$ 1, 5, and 10 kg m$^{-2}$ accounts for 43%, 76%, and 94%, respectively, all grids for Globbiomass, and 51%, 83%, and 96%, respectively, for GEOCARBON (Fig. 1b). Figure 1b clearly indicates that the probability distribution of AGB availability for Globbiomass reflects larger values relative to that of GEOCARBON in the range of AGB $\leq$ 10 kg m$^{-2}$. Overall, Globbiomass
indicates approximately $1.35\times$ more AGB than GEOCARBON; however, in the range of AGB $\geq 25$ kg m$^{-2}$, the AGB for GEOCARBON is greater than that for Globbiomass. This is partly associated with a tendency for the AGB estimates in GEOCARBON to be higher than those in previous studies of dense forest areas, mainly tropical evergreen broadleaf forest (Avitabile et al., 2016) (Figs. S1 and S2).

The emission factor (EF) and burning efficiency (BE), which are related to the nature of the flammable materials that
comprise the AGB and control the BB emissions (Eqs. 1 and 2), are defined by the LCC. To quantify the differences between the two LCC maps used (Fig. S2), we calculated global area totals for three vegetation classes: forest, shrub/savanna/grass, and crop, as defined in the GLC2000 and MCD12Q1 LCCs (Table 2). The LCC data from 2009 were used for MCD12Q1 in this comparison. The forest area in GLC2000 was $55.8 \times 10^6$ km$^2$, 199% more than that in MCD12Q1 ($28.0 \times 10^6$ km$^2$); the area of shrub/savanna/grass in GLC200 is $56.4 \times 10^6$ km$^2$, 43% less than MCD12Q1 ($98.6 \times 10^6$ km$^2$); that of crop in GLC2000
was $28.2 \times 10^6$ km$^2$, 181% more than MCD12Q1 ($15.6 \times 10^6$ km$^2$). At the global scale, it is noteworthy that there are large differences in the area totals of the vegetation classes between the two products; e.g., GLC2000 possesses larger forest areas, whereas MCD12Q1 has more shrub/savanna/grass. Giri et al. (2005) found that the spatial distribution of vegetation in eight LCC classes shows agreement of 59.5% between the GLC2000 and MCD12Q1 products, and the discrepancies between them occur in southern Siberia, the Sahel region, southeastern Brazil, Southern Australia, and on the Tibetan plateau.

These differences in AGB and LCC distributions can be expected to result in differing BB CO emissions estimates among the four AGB/LCC scenarios. We first calculated annual average of BB CO emissions (g CO m$^{-2}$ yr$^{-1}$) for 2009–2015 for each estimate (Fig. 2). The four estimates and their ensemble average show similar spatial pattern of BB emissions, at least at the globe scale: large emissions in tropical regions over northern and southern tropical Africa, tropical Asia, and Oceania, as well as the southern Amazon, boreal Eurasia, and northern and southeastern regions of North America. However, the magnitudes of the emissions differ among the estimates. The standard deviation among the four BB emissions estimates increases as BB

emissions increase, particularly in northern and southern tropical Africa.

  Global monthly BB CO emissions exhibit seasonal variability with peaks during the boreal summer for all estimates (Fig. 3a). In addition, the GlcGlob and McdGlob estimates show second peaks during the austral summer, whereas the GlcGeoc and McdGeoc estimates show no well-defined peaks. This suggests that the seasonal variability among the BB estimates will change

depending on the AGB map, and that the emission totals vary with both the AGB and LCC maps. The annual emission totals vary significantly depending on which BB estimates are used. The mean annual BB emissions with their standard deviation are $526 \pm 53$, $219 \pm 35$, $624 \pm 57$, and $293 \pm 44$ Tg CO yr$^{-1}$, for GlcGlob, GlcGeoc, McdGlob, and McdGeoc, respectively. The EsmAve displays moderate seasonal variability in global monthly BB CO emissions and the mean annual emissions of $415 \pm 47$ Tg CO yr$^{-1}$. Using data from over 370 publications, Andreae (2019) obtained CO emissions in the range 390–1,210 Tg CO

yr$^{-1}$ from open vegetation fires and 181–196 Tg CO yr$^{-1}$ from indoor biofuel use. Our study does not take separate account of emissions from biofuel use, which could result in a slight underestimation of total emissions from BB. The annual CO emissions from BB reported in Andreae (2019) span a wide range, and our emissions estimates based on GlcGlob, McdGlob, and EsmAve fall within this range. However, our estimates based on GlcGeoc and McdGeoc fall substantially below this range. On the other hand, GFED4.1s and GFASv1.2 shows the mean annual BB emissions less than 350 Tg CO yr$^{-1}$ (Table 4) for

2009–2015, the differences percentages from them are larger for McdGlob (+88%) than GlcGeoc (-37%).

  To evaluate the sensitivity of BB emissions estimates to land surface information at the regional scale, we next compared seasonal variability in BB CO emissions from the four estimates and their ensemble average over southern tropical Africa (see black rectangle in Fig. 2f) (Fig. 3b). This region is situated in a complicated transition zone containing forest, -savanna, and -bare ground, and with few local studies, and this has led to poor quality land surface information and a high degree

of variability among the datasets (Bouvet et al., 2018). All estimates reveal fire emissions from May to October. The annual emissions estimates based on the four AGB/LCC scenarios range over a factor of 4 from 36 to 146 Tg CO yr$^{-1}$ among the estimates. In southern tropical Africa, McdGlob has the highest AGB of 2,704 g m$^{-2}$ for shrub/savanna/grass with higher BE (Table 3), and this rich supply of flammable fuel leads to the highest CO emissions. For the McdGlob estimate, the AGB of 3.1% for shrub/savanna/grass is burned annually and is converted into emissions. By contrast, GlcGeoc has the low-

est AGB for shrub/savanna/grass (28 g m$^{-2}$) and this results in the lowest CO emissions. This large difference in AGB for shrub/savanna/grass between McdGlob and GlcGeoc, that is between Globbimass and GEOCARBON, over the southern tropical Africa is not surprising. Globbimass is an AGB product based partly on the retrieval of biomass from airborne L-band SAR data, which interact with the Earth's surface; the data are sensitive to forest vegetation's primary and secondary branches and stems (Lucas et al., 2010; Carreiras et al., 2012). Although the L-band SAR data are influenced by moisture on vegetation

and soil, and exhibit saturation in dense forest, successful biomass estimation is promising over shrub and savanna lands where surface moisture is relatively low. GEOCARBON is, on the other hand, primarily a global forest AGB map, which is derived from two pan-tropical AGB maps with calibration using reference field measurement datasets. The uneven distribution of AGB reference datasets toward forest vegetation types leads to lower AGB estimates than other previous studies of Central America and mostly dry vegetation areas in Africa (Avitabile et al., 2016). The high discrepancy of CO emissions for this area hence resulted from the difference in AGBs for forest and shrub/savanna/grass among the four scenarios.

In order to investigate the BB emissions estimates further, mean annual BB CO emissions over 7 years in the 14 global regions (Fig. A1) are listed in Table 4, along with those of GFED4.1s and GFASv1.2. There was not much difference in mean annual BB CO emissions between GFED4.1s and GFASv1.2 for almost all the regions and for the global total, although the Southern Hemisphere Africa (SHAF) and Boreal Asia (BOAS) showed some differences. Mean annual emissions of the four BB estimates and EsmAve over the BOAS region (29.4–58.8 Tg CO yr$^{-1}$) fall between those of GFED4.1s and GFASv1.2 (29.2 and 67.2 Tg CO yr$^{-1}$), whereas the variations among the four BB estimates over the SHAF region (38.9–164.5 Tg) are substantially larger than the range between GFED4.1s and GFASv1.2 (90.6 and 61.1 Tg CO yr$^{-1}$). Large variations among the four BB emissions estimates are also found in Northern Hemisphere Africa (NHAF) (range between 16.6 and 127.1 Tg CO yr$^{-1}$). The large variations can be attributed to the low confidence of land surface information over the NHAF and SHAF regions, as mentioned in Fig. 3b. The mean annual BB CO emissions of the four BB estimates were consistently lower than those of GFED4.1s and GFASv1.2 in Equatorial Asia (EQAS) and in Boreal North America (BONA), and higher in Temperate North America (TENA) and in Southern Hemisphere South America (SHSA). It is not straightforward to consistently interpret these different patterns in BB emissions over the regions relative to GFED4.1s and GFASv1.2, at least for GFED4.1s, because this study estimated BB CO emissions using a similar burned area method and MODIS fire product to GFED4.1s. A potential reason for the differences might be differences in the AGB datasets used in the estimates. This study uses the AGB datasets from Globbimass and GEOCARBON, while GFED4.1s uses the CASA model for simulation. Although the AGB used in GFED4.1s is adjusted to match GEOCARBON at the biome level, carbon allocation and variation in the vegetation carbon pool (including the impact of fire-induced mortality and turnover) are also represented in the CASA model, partly using the satellite-derived instantaneous tree mortality information (van der Werf et al., 2017). These detailed processes using the vegetation carbon pool are not used in our estimates. These different representations of AGB could result in the differing regional BB CO emissions. The different spatiotemporal resolutions of the estimates is another potential cause. This study estimated monthly BB emissions at 500 m spatial resolution, whereas GFED4.1s and GFASv1.2 estimated 3 hourly emissions at 0.25 grid resolution and daily ones at 0.1 grid resolution, respectively. A coarse spatiotemporal resolution reduces the variability of land cover types and flammable fuel consumption, leading to differing amounts of BB emissions (Wees and Werf, 2019).

## 3.2 Comparisons of modeled atmospheric CO fields

As an alternative approach, we compared modeled CO fields using each BB emission estimate. Variability in atmospheric CO concentrations was simulated using NICAM-TM with surface flux information including the four BB emissions estimates. Observed and simulated daily time series at the three ground-based observation sites, BKT, ETL, and MNM are shown in

Fig. 4. The observations at the BKT site in Indonesia and the ETL site in Canada may be subject to recurrent fire events. In particular, daily average CO concentrations exceeding 1,000 ppb were frequently observed at the BKT site for 2014 and 2015, leading to a mean and standard deviation of $660.2 \pm 707.5$ ppb for fire months and $153.3 \pm 53.2$ ppb for no fire months (Table 5). The mean concentrations for no fire and fire months (130.6 and 163.7 ppb, respectively) were not significantly different at the ETL site. The variability in observed daily atmospheric CO concentrations for both no fire and fire months was moderately captured by simulations at both sites, regardless of which BB emissions estimate was used. Indeed, there is no clear difference in correlation coefficient and MIA among the simulations using the four BB emissions estimates for no fire months. However, the simulated variability during fire events for fire months is generally weakened relative to the observations, and the impact of different BB emissions estimates on CO concentrations appears in the amplitude of the simulated variability. The variability simulated using McdGlob shows highest concentrations, and that for GlcGeoc shows the lowest ones. For fire months, deviation of the simulated variability from the observations is apparent for simulations based on BB estimates that used the GEOCARBON AGB map, lower correlation coefficients at the BKT site, and lower MIA at the ETL site.

These temporal characteristics of daily variability in simulated atmospheric CO concentrations can, however, be largely dominated by atmospheric transport processes simulated using NICAM-TM, not by the BB emissions estimates, because the BB emissions estimates have a monthly resolution. To reduce this inconsistency in the temporal resolutions of the observations and the BB emissions estimates, the observed and simulated daily time series were averaged to monthly ones (Fig. 5). For no fire months, although simulated atmospheric concentrations for the four BB emissions estimates show large variations at both the BKT and ETL sites, they display monotonic trends along identity lines to the observations. For fire months, simulated variability displays opposing trends at the two sites: underestimations of atmospheric concentrations at the BKT site and overestimations at the ETL site for all BB emissions estimates. These comparisons reveal a difficulty faced by the simulations in reproducing the ground-based observations of higher CO concentrations generated by sudden BB emissions from intermittent fire events.

At the MNM site in Japan, with no local fire events, there was no clear difference among the BB emissions estimates in terms of the correlation coefficient, mean absolute error, and MIA (Fig. 4 and Table 5). Differences in the mean CO concentration among the simulations at the MNM site (11.2 ppb) were smaller than those at BKT (18.3 ppb) and ETL (22.0 ppb) for the no fire months. However, the difference at the MNM site implies that differences in BB emissions estimates can even contribute to variability in the background atmospheric CO concentration, even though CO has a relatively short lifespan in the atmosphere of weeks to months.

To extend the comparison over the regional scale, the global distributions of XCO (ppb) were averaged for 2013–2015, for the MOPITT observations and the simulations using the four BB emissions estimates and their ensemble average (Fig. 6). All of the results in Fig. 6 show strong spatial variations in XCO. Higher concentrations of XCO are found over tropical regions, southeastern North America, boreal Eurasia, and southeast Asia in the MOPITT observations. These regions are consistent with the areas with large BB emissions, as shown in Fig. 2. Lower XCO concentrations are found over the oceans in the southern hemisphere in the MOPITT observations. These global distributions of XCO are represented in the simulations from all of

the BB emissions estimates, but the mean XCO concentrations at the regional scale differ in the simulations among the BB emissions estimates.

Figure 7 shows monthly mean XCO and the root mean square error (RMSE, ppb) between the observed and simulated XCO fields over six selected areas: West Coast of the United States of America (WCA), Eastern Siberia (ESB), the Amazon (AMZ), South Asia (SAS), Central Africa (CAF), and the Sumatra and Borneo Islands (SBI), which are shown in the red rectangle in Fig. 6a. Over the WCA and ESB areas, the monthly mean observed XCO shows little seasonality, with standard deviations of 11.5 and 10.4 ppb (Fig. 7a and b; Table 6). During the approximately three months of the year with higher XCO concentrations,

the mean observed XCO concentrations over both areas increases by approximately 11–17% relative to the other months. The XCO values simulated using GlcGlob or EsmAve largely reproduce the observed seasonality, but those from GlcGeoc show less seasonality, resulting in a higher RMSE and lower MIA. Underestimations of peak concentrations and seasonal variability in simulations with GlcGeoc are also apparent over the AMZ areas, with moderate seasonality and standard deviations of 23.5 ppb, and also over the CAF and SBI areas, with large seasonality and standard deviations of 28.7 and 29.5 ppb, respectively.

The XCO values simulated using McdGlob, on the other hand, show overestimates of BB emissions during the fire seasons, in particular, over the ESB and CAF areas, whereas the abrupt increase in XCO (195.4 ppb) in October 2015 over the SBI area is well represented by the monthly mean value of 185.9 ppb. The values simulated using GlcGlob, McdGeoc, and EsmAve are between those from McdGlob and GlcGeoc, and they moderately recreate the observed variability, except over the SAS area. The simulated XCO values over the SAS area, where contributions from fossil fuels to CO emissions prevail over BB

emissions (Yarragunta et al., 2021), were approximately 20 ppb higher than the observations over the whole period, regardless of which BB emissions estimate was used. This suggests that representation of CO emissions remains insufficient over the SAS area, not only for BB emissions but also other emission sources. Values of MIA show that the mean value over the five areas (excluding SAS) was better in the simulations based on GlcGlob, with a value of 0.66 (0.58 to 0.71), whereas those derived using GlcGeoc, McdGlob, and McdGeoc, were 0.60 (0.54 to 0.72), 0.61 (0.56 to 0.68), and 0.63 (0.55 to 0.75), respectively.

Additionally, EsmAve exhibited the highest value of 0.68 (0.59 to 0.74), although differences in the mean values of MIA from the simulations using other BB emissions estimates were not large.

## 4   Discussion

BB emissions are an important contributor to atmospheric greenhouse gases and aerosols, yet uncertainty with respect to regional and interannual variability remains due to our limited understanding of the underlying mechanisms and lack of data

related to this variability. Accurate and detailed information regarding AGB and LCC is essential to estimates of BB emissions from wildfires using the bottom-up approach. Improvements in satellite sensors, ground surface observations, digital image processing techniques, and retrieval algorithms have contributed towards reducing the uncertainties associated with AGB and LCC mapping (e.g., Goetz et al., 2009; Clerici et al., 2017). Nevertheless, datasets prepared using different data sources, classification schemes, and methodologies generate discrepancies in the AGB and LCC distributions among the products (Fig. 1;

Table 2) as has been discussed previously (e.g., Giri et al., 2005).

This study tested combinations of two sources of AGB data, Globbiomass and GEOCARBON, and two sources of LCC data, GLC2000 and MCD12Q1, and used the same burned area satellite data to estimate BB CO emissions. Although the EF and BE parameters remained the same in our estimates, our analysis showed large discrepancies in annual mean CO emissions, with a factor of approximately 3 (219 – 624 Tg CO $yr^{-1}$) separating the four BB emissions estimates. Using AGB data from
Globbiomass and GEOCARBON, we showed that the magnitude of AGB from Globbiomass tends to be approximately 35% larger than that from GEOCARBON, leading to the resulting BB emissions estimates based on Globbiomass being more than twice those made using GEOCARBON over the globe. Furthermore, our comparison of the LCC data showed that the global area totals for the forest class in GLC200 were approximately twice those for MCD12Q1, while those for shrub/savanna/grass in GLC2000 were approximately half those in MCD12Q1. As burning efficiencies for shrub/savanna/grass are greater than those for forests (Table A1 and A2), the BB emissions based on MCD12Q1, with its larger area totals for shrub/savanna/grass,
tend to be higher than those based on GLC2000. These results indicate that the estimates of BB emissions are highly sensitive to the AGB and LCC data, and thus the AGB and LCC data used could be the primary drivers of uncertainty in the estimates of BB emissions. In addition, because adequately accurate distributions of AGB and LCC are still unavailable, an independent approach is needed to evaluate the estimate of BB emissions.

Variability in atmospheric CO concentrations simulated using an atmospheric tracer transport model and the BB emissions and other emission inventories were compared with ground-based and satellite observations to act as the independent evaluation of the BB emissions estimates. We did not take account of errors introduced by the observational processes or errors in the transport model and the other emission inventories, but we consider that our analysis is a useful way to study the relative differences among the BB emissions estimates and approximate changes in the simulated atmospheric concentrations. Extend-
ing this analysis to ground-based observations of the impact of intermittent fire events at the local scale was more challenging due to the coarse resolution of the available BB emissions estimates and the atmospheric tracer transport model, which weakens temporal and spatial variability in the simulated atmospheric CO concentrations. Abrupt variability in atmospheric CO concentrations recorded in the ground-based observations for fire months were indeed represented with the variations that are attenuated in the higher CO concentrations (Fig. 4). Relatively small differences among the BB emissions estimates from the
ground-based observation sites (Table 5) may be attributed to the loss of information related to the high-frequency variability in the simulated atmospheric CO concentration. We need to recognize that a global transport model with a horizontal resolution of about 220 km is insufficient to quantify local BB emissions accurately. The attenuation in the simulation can be moderately improved by including daily variability in the BB emissions, especially for surface observations with high levels of biomass burning, using atmospheric transport simulations with a high spatial resolution (e.g., Mu et al., 2011).

At the global scale, comparison with satellite observations suggests that the XCO variability simulated using the AGB data from Globbiomass and the LCC data from GLC2000, and that from the ensemble average of the four emissions estimates, provides a better representation of the temporal and spatial variability in observed XCO during fire seasons than that achievable using other combinations of the AGB and LCC data (Figs 6 and 7; Table 6). The GlcGlob and EsmAve estimates yield global BB emissions of $526\pm53$ and $415\pm47$ Tg CO $yr^{-1}$. The total CO emissions of GlcGlob are slightly higher than those reported
by Hooghiemstra et al. (2011), who found total emissions of $400\pm88$ and $482\pm68$ Tg CO $yr^{-1}$ for 2003 and 2004, respectively,

using a data assimilation to surface observations, whereas the EsmAve exhibits total CO emissions close to the estimates by Hooghiemstra et al. (2011) for 2003. The corresponding mean emissions for 2009 and 2015 obtained from GFED4.1s and GFASv1.2 were 331 and 350 Tg CO yr$^{-1}$, respectively, which are approximately 37% and 34% lower than GlcGlob, and approximately 20% and 16% lower than EsmAve.

Note that our analysis is not a guarantee of the validity of the AGB and LCC data used, and we do not intend to argue which of the AGB and LCC datasets are better than others. As CO EFs remain uncertain, due mainly to the difficulty in treatment of emissions from residual smoldering combustion (Andreae, 2019), the estimated BB emissions can vary according to the EF used and depend on the selection of the fire class confidence in the fire mask data. Further, this study used the EF classified based on LCC and the location of the objective grid in a region (Tables A1 and A2), but the magnitudes of EF can vary with

the plant species even for the same LCC type. For example, the CO EFs for crop straws vary between 27.2 and 46.9 g kg$^{-1}$ dry matter among rice, wheat, and barley under dry conditions owing to different lengths of combustion time with smoldering, and these EFs change easily with moisture levels in crop residues (Hayashi et al., 2014). Such detailed variability of EF can only be accounted for by estimating BB emissions at a finer resolution and by using a finer vegetation classification map. Additionally, one limitation of the current study of BB estimates is that it does not include a scheme to inherit the amount of

AGB that remained unburned in the previous year. Although continuous variations in AGB over multiyear periods, and the impact of these variations on BB emissions, can be simulated by coupling the system to a terrestrial biosphere model, this work remains incomplete. Finally, to improve our currently limited ability to estimate BB emissions, we are calling for additional independent approaches and data evaluation to help increase our understanding of their characteristics.

## 5    Conclusions

This study used the burned area method in bottom-up approaches to estimate spatiotemporal variations in global BB CO emissions based on AGB and LCC land surface information and burned area data. Regarding the land surface information, we tested two AGB datasets (Globbiomass and GEOCARBON) to evaluate the sensitivity of BB emissions estimates to these different datasets. Preliminary comparisons of the AGB and LCC datasets showed substantial differences among them. The spatial distribution of AGB was highly correlated between Globbiomass and GEOCARBON, but the former contained AGB

values that were larger than the latter by a factor of 1.35. The global area total for forest in GLC2000 was 199% more than that in MCD12Q1, but was 43% less than for shrub/savanna/grass. By combining these AGB and LCC data with the burned area data, four BB emissions estimates (i.e., GlcGlob, GlcGeoc, McdGlob, and McdGeoc) were derived using the burned area method.

     We began by comparing the seasonal variability of the BB emissions estimates over the regional and global scales. This

comparison showed that BB emissions increase as the amount of AGB for shrub/savanna/grass increases over the corresponding burned area. Our estimates of the mean annual BB emissions resulted in a large divergence among the estimates; i.e., $526\pm53$, $219\pm35$, $624\pm57$, and $293\pm44$ Tg CO yr$^{-1}$ for GlcGlob, GlcGeoc, McdGlob, and McdGeoc, respectively, and their ensemble average EsmAve was $415\pm47$ Tg CO yr$^{-1}$. Using the BB emissions estimates, variability in atmospheric CO concentrations

was simulated using NICAM-TM with other emissions sources (i.e., fossil fuel and biogenic emissions) as inputs. We evaluated our results against independent ground-based (WDCGG network) and satellite (MOPITT) CO observations. Comparison with data from the ground-based sites indicated that all BB emissions estimates represent local fire events, but underestimation of BB emissions was particularly apparent for intense fires at the BKT site in Indonesia. Explicit differences in the simulated CO concentrations among the BB emissions estimates were found in comparison with the satellite observations at the regional scale. In our simulations, the XCO variability simulated using the GlcGlob estimates or the EsmAve was the most consistent with the satellite observations at the regional and global scales.

This study has confirmed that BB emissions estimates are sensitive to the land surface information on which they are based. Furthermore, although it is clear that there are significant differences among the various land surface information products currently available, the quantitative evaluation of these differences remains difficult because of the limited coverage of surface observations. One approach to addressing this limitation would be the commissioning of future satellite missions carrying higher-resolution onboard sensors.

*Acknowledgements.* This study was supported by the GOSAT-2 project at the National Institute for Environmental Studies (NIES) and the Climate Change Adaptation Research Program of NIES. The simulations were completed using the supercomputer (NEC SX-Aurora TSUBASA) at NIES. We thank the EDGAR, ESA, GEOCARBON, GLC2000, Globbiomass, JRA-55, MODIS, MOPITT, and WDCGG teams, as well as Dr. Akihiko Ito (NIES), for providing the data used in this study. BKT is operated by the Indonesian Agency for Meteorology Climatology and Geophysics (BMKG) with support from the WMO/GAW Quality Assurance/Science Activity Centre Switzerland (QA/SAC-CH).

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

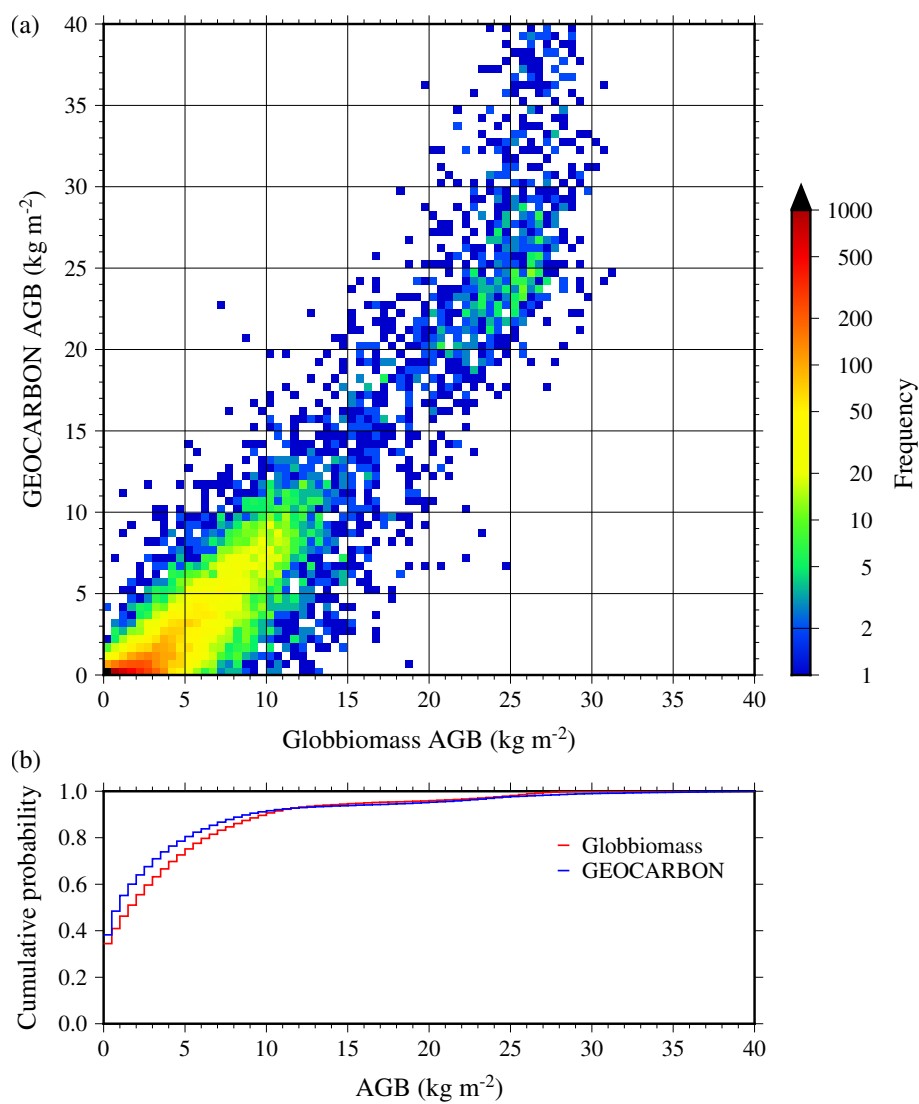

**Figure 1.** Comparisons of (a) AGB for 2009 based on Globbiomass and GEOCARBON and (b) their histograms.

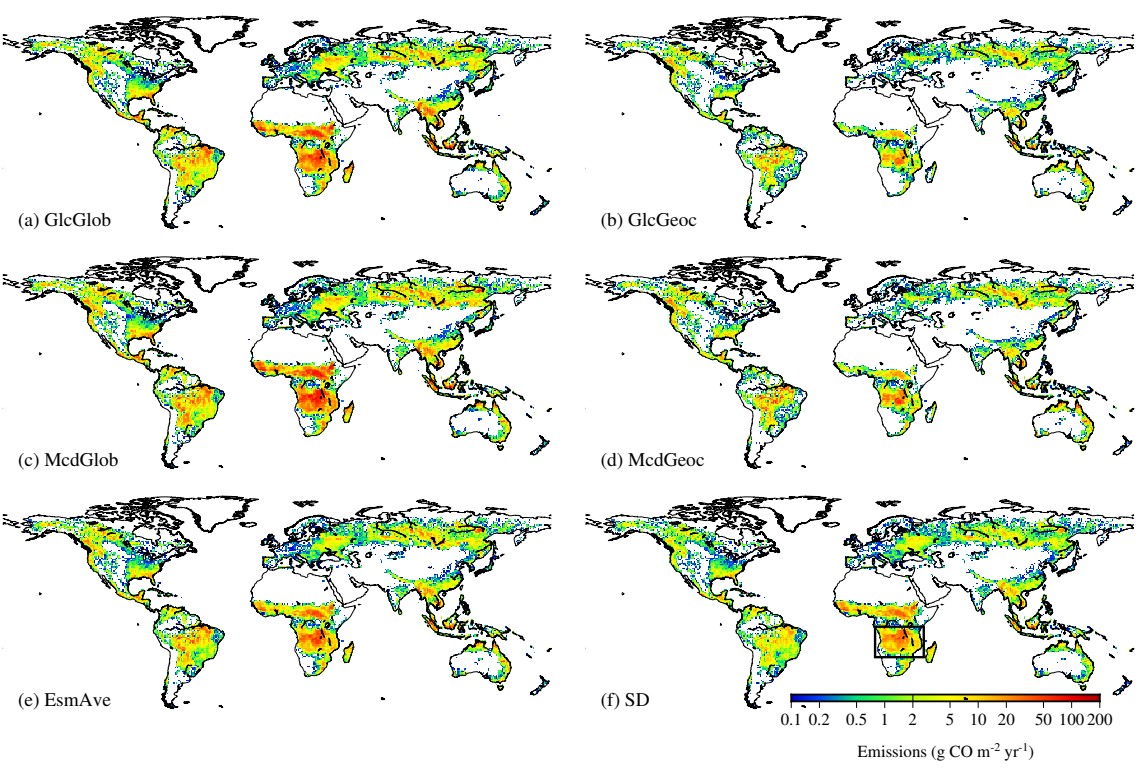

**Figure 2.** Spatial distributions of the annual average of CO emissions (g CO m$^{-2}$ yr$^{-1}$) for (a) GlcGlob, (b) GlcGeoc, (c) McdGlob, (d) McdGeoc, and (e) EsmAve and (f) their standard deviation based on four BB emissions estimates over the period 2009–2015.

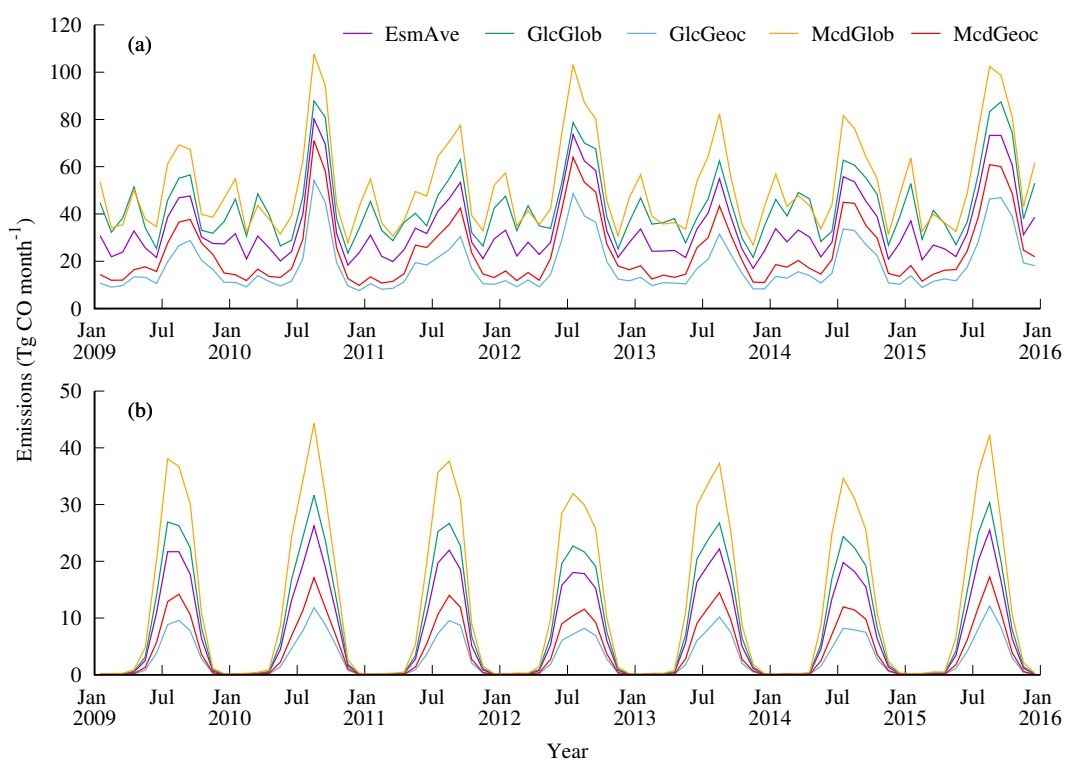

**Figure 3.** Monthly CO emissions (Tg CO month$^{-1}$) for EsmAve (purple line), GlcGlob (green line), GlcGeoc (blue line), McdGlob (orange line), and McdGeoc (red line) over (a) the globe and (b) the Southern Africa region within the black rectangle shown in Fig. 2f.


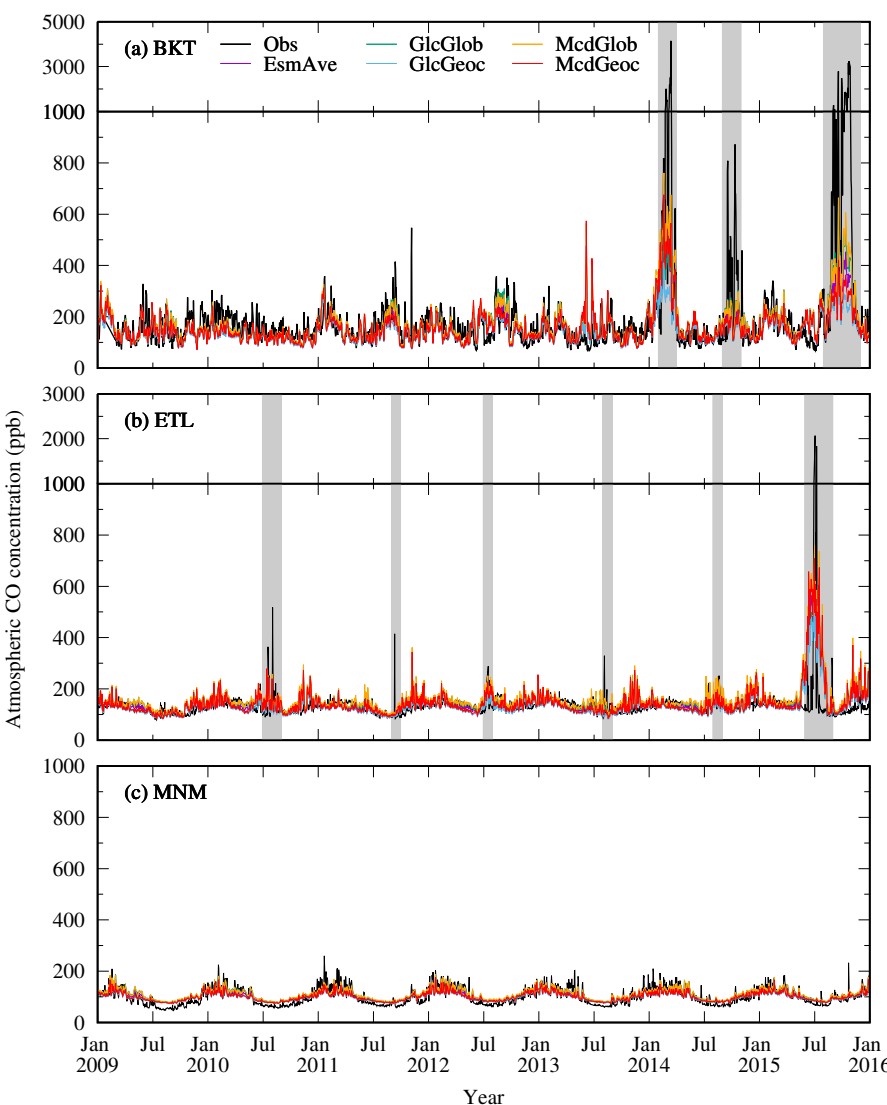

**Figure 4.** Daily atmospheric CO concentration variations (ppb) at the three ground-based observation stations, (a) Bukit Kototabang (BKT), (b) East Trout Lake (ETL), and (c) Minamitorishima (MNM) for 2009–2015. The grey shading for the BKT and ETL sites indicates the fire months identified using the standardized anomaly (Equation 8).

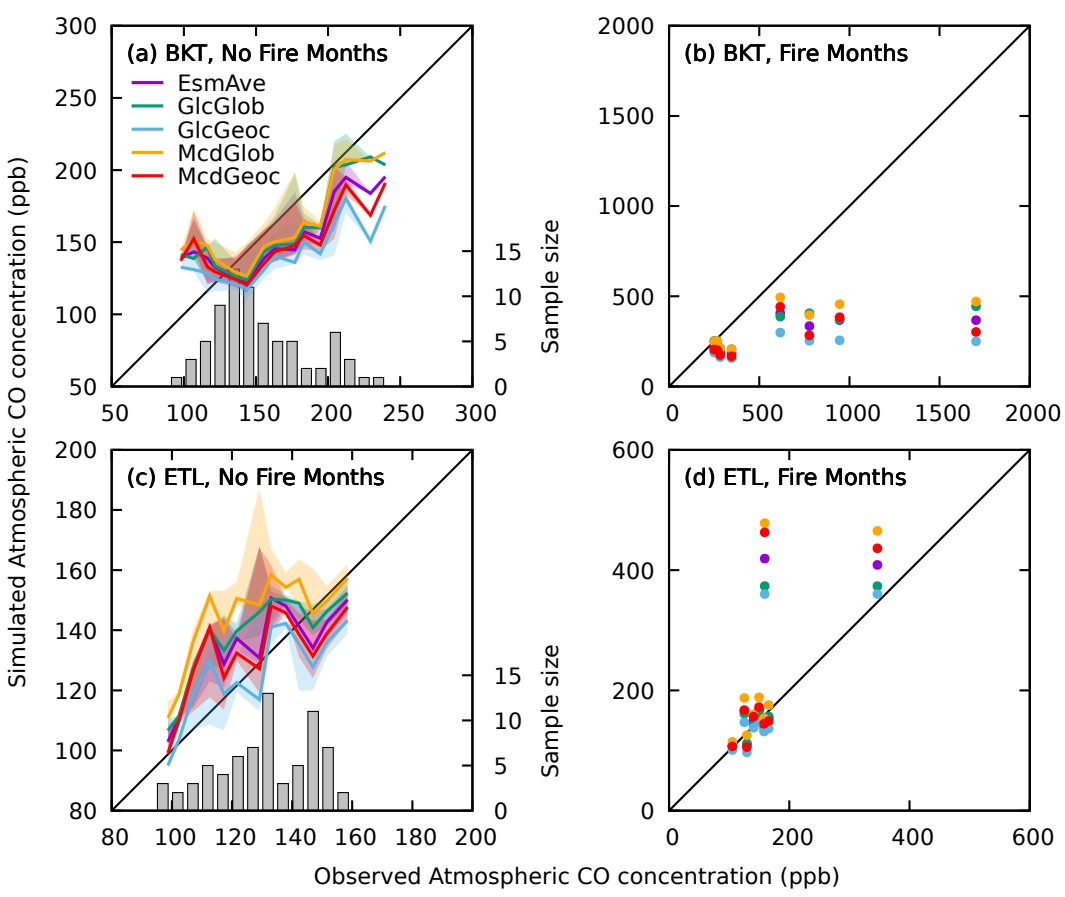

**Figure 5.** Comparisons of observed and simulated monthly atmospheric CO concentration (ppb) over the period 2009–2015. The conditional quantile plots for no fire months at (a) BKT and (c) ETL. The scatter plots for fire months at BKT (b) and (d) ETL.

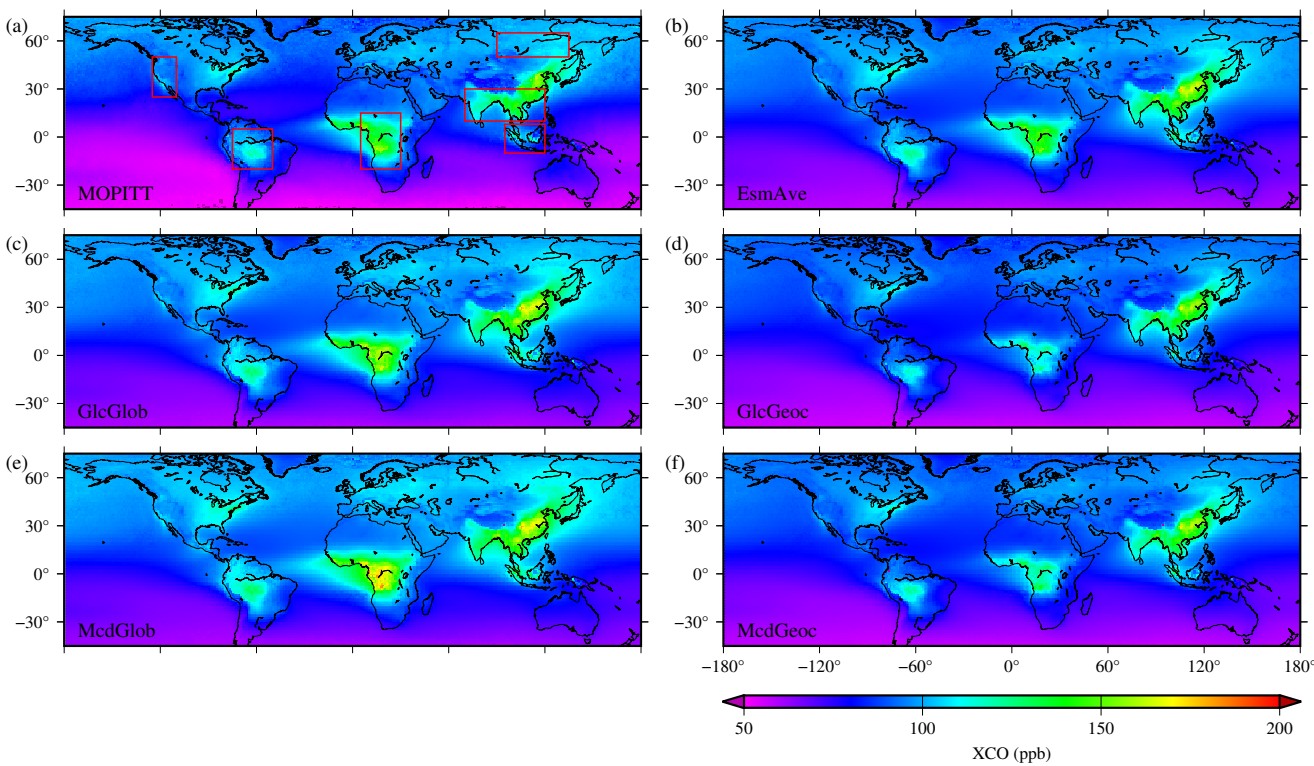

**Figure 6.** Spatial distributions of mean XCO (ppb) between 2013 and 2015 for (a) MOPITT Level 3 product and the simulations using (b) EsmAve, (c) GlcGlob, (d) GlcGeoc, (e) McdGlob, and (f) McdGeoc.

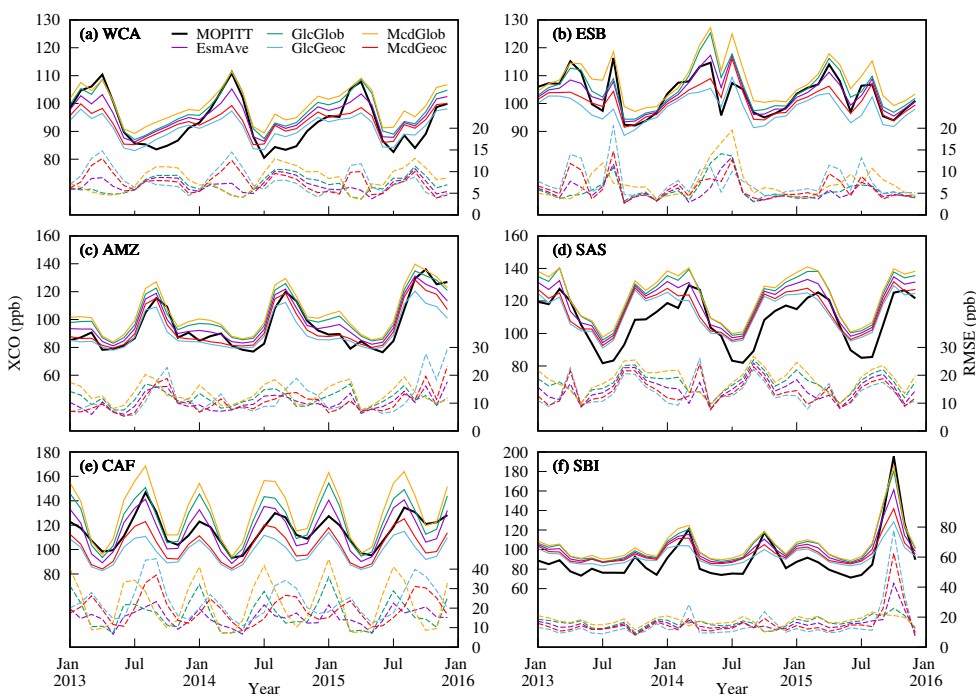

**Figure 7.** Monthly mean XCO variations (ppb; solid) and RMSE (ppb; dashed) between observed and simulated fields over the six areas: (a) WCA, (b) ESB, (c) AMZ, (d) SAS, (e) CAF, and (f) SBI for 2013–2015.

**Table 1.** BB emissions estimates and the LCC and AGB data used.

| Product | LCC map | AGB map |
|---------|---------|---------|
| GlcGlob | GLC2000 | Globbiomasss |
| GlcGeoc | GLC2000 | GEOCARBON |
| McdGlob | MCD12Q1 | Globbiomass |
| McdGeoc | MCD12Q1 | GEOCARBON |


**Table 2.** Global area totals ($10^6$ km$^2$) for forest, shrub/savanna/grass, and crop land[1]in GLC2000 and MCD12Q1.

| Type | GLC2000 | MCD12Q1 |
|------|---------|---------|
| Forest | 55.8 | 28.0 |
| Shrub/Savanna/Grass | 56.4 | 98.6 |
| Crop | 28.2 | 15.6 |

[1]Forest: Tree Cover, broadleaved, evergreen, Tree Cover, broadleaved, deciduous, closed and open, Tree Cover, needle-leaved, evergreen and deciduous, Tree Cover, mixed leaf type, and Mosaic: Tree Cover, Other natural vegetation for GLC2000; Evergreen Needleleaf Forest, Evergreen Broadleaf Forest, Deciduous Needleleaf Forest, Deciduous Broadleaf Forests, and Mixed Forests for MCD12Q1. Shrub/Savanna/Grass: Shrub Cover, closed-open, evergreen and deciduous, Herbaceous Cover, closed-open, and Sparse herbaceous or sparse shrub cover for GLC2000; Closed Shrublands, Open Shrublands, Woody Savannas, savannas, and Grassland for MCD12Q1. Crop: Cultivated and managed areas, Mosaic: Cropland, Tree Cover, Other natural vegetation, and Mosaic: Cropland, Shrub and/or grass cover for GLC2000; Croplands and Cropland/Natural Vegetation Mosaics for MCD12Q1.

**Table 3.** Mean AGB (g m$^{-2}$) from the four BB emissions estimates for forest and shrub/savanna/grass over southern tropical Africa (Fig. 2b). Numbers in parentheses are the annual AGB decrement (%) caused by fires.

| Type | GlcGlob | GlcGeoc | McdGlob | McdGeoc |
|---|---|---|---|---|
| Forest | 3,567 (1.0) | 2,749 (0.9) | 1,896 (0.7) | 1,925 (0.7) |
| Shrub/Savanna/Grass | 738 (2.2) | 28 (1.3) | 2,704 (3.1) | 893 (3.2) |

**Table 4.** Mean annual BB CO emissions (Tg yr$^{-1}$) from the GFED4.1s, GFASv1.2, and this study estimates over the 14 global regions (see Fig. A1) between 2009 and 2015.

| Region | GFED4.1s | GFASv1.2 | EsmAve | GlcGlob | GlcGeoc | McdGlob | McdGeoc |
|--------|----------|----------|--------|---------|---------|---------|---------|
| BONA | 21.5 | 34.5 | 16.9 | 17.0 | 12.7 | 21.6 | 16.4 |
| TENA | 3.3 | 7.0 | 16.3 | 17.5 | 9.0 | 24.3 | 14.3 |
| CEAM | 4.4 | 5.2 | 11.0 | 15.5 | 4.8 | 16.5 | 7.1 |
| NHSA | 3.6 | 3.7 | 7.4 | 12.0 | 2.6 | 11.5 | 3.6 |
| SHSA | 39.3 | 33.2 | 68.5 | 78.1 | 48.5 | 84.0 | 63.3 |
| EURO | 1.1 | 1.3 | 2.7 | 3.5 | 1.2 | 4.3 | 2.0 |
| MIDE | 0.3 | 1.5 | 0.5 | 0.6 | 0.2 | 1.0 | 0.3 |
| NHAF | 52.4 | 42.9 | 67.0 | 104.5 | 16.6 | 127.1 | 20.0 |
| SHAF | 90.6 | 61.1 | 94.6 | 119.9 | 38.9 | 164.5 | 55.0 |
| BOAS | 29.2 | 67.2 | 43.5 | 48.3 | 29.4 | 58.5 | 37.5 |
| CEAS | 10.4 | 12.8 | 13.5 | 20.4 | 4.8 | 21.8 | 7.2 |
| SEAS | 19.1 | 20.4 | 26.4 | 40.8 | 14.6 | 33.4 | 16.9 |
| EQAS | 40.4 | 39.5 | 25.6 | 33.1 | 17.2 | 31.8 | 20.3 |
| AUST | 14.8 | 19.6 | 21.2 | 14.5 | 18.2 | 23.1 | 29.1 |
| Total | 331 | 350 | 415 | 526 | 219 | 624 | 293 |

**Table 5.** Statistics comparing observed and simulated time series of daily atmospheric CO concentrations at the BKT, ETL, and MNM sites between 2009 and 2015.

| Statistics | No Fire Months | | | Fire Months | |
|---|---|---|---|---|---|
| | BKT | ETL | MNM | BKT | ETL |
| No. of Observations | 2,236 | 2,053 | 2,511 | 237 | 226 |
| Mean (ppb) | | | | | |
| Observations | 153.3 | 130.6 | 101.3 | 660.2 | 163.7 |
| EsmAve | 148.4 | 140.0 | 104.5 | 287.3 | 198.0 |
| GlcGlob | 152.6 | 142.8 | 108.0 | 317.1 | 187.4 |
| GlcGeoc | 137.9 | 129.2 | 98.9 | 218.8 | 176.2 |
| McdGlob | 156.2 | 151.2 | 110.1 | 342.7 | 222.0 |
| McdGeoc | 146.9 | 136.8 | 101.1 | 269.9 | 206.3 |
| Standard deviation (ppb) | | | | | |
| Observations | 53.2 | 19.7 | 32.1 | 707.5 | 196.4 |
| EsmAve | 44.2 | 27.9 | 18.2 | 104.5 | 126.6 |
| GlcGlob | 47.1 | 26.4 | 19.8 | 113.2 | 109.5 |
| GlcGeoc | 37.8 | 26.0 | 16.4 | 61.2 | 108.3 |
| McdGlob | 49.1 | 30.7 | 20.2 | 142.7 | 146.8 |
| McdGeoc | 45.1 | 30.1 | 16.9 | 112.1 | 142.4 |
| Mean absolute error (ppb) | | | | | |
| EsmAve | 35.5 | 19.8 | 15.0 | 407.2 | 96.8 |
| GlcGlob | 35.3 | 19.3 | 15.5 | 388.3 | 89.5 |
| GlcGeoc | 36.5 | 19.1 | 15.3 | 456.2 | 86.3 |
| McdGlob | 36.1 | 25.0 | 16.2 | 376.1 | 112.0 |
| McdGeoc | 36.8 | 21.4 | 15.2 | 423.0 | 103.8 |
| Correlation coefficient | | | | | |
| EsmAve | 0.51 | 0.38 | 0.88 | 0.56 | 0.31 |
| GlcGlob | 0.54 | 0.43 | 0.87 | 0.61 | 0.31 |
| GlcGeoc | 0.49 | 0.41 | 0.87 | 0.46 | 0.31 |
| McdGlob | 0.51 | 0.35 | 0.88 | 0.60 | 0.31 |
| McdGeoc | 0.46 | 0.34 | 0.87 | 0.45 | 0.31 |
| Modified index of agreement | | | | | |
| EsmAve | 0.54 | 0.45 | 0.64 | 0.54 | 0.35 |
| GlcGlob | 0.55 | 0.47 | 0.65 | 0.54 | 0.41 |
| GlcGeoc | 0.51 | 0.46 | 0.62 | 0.52 | 0.29 |
| McdGlob | 0.54 | 0.40 | 0.64 | 0.54 | 0.44 |
| McdGeoc | 0.52 | 0.42 | 0.63 | 0.53 | 0.31 |

**Table 6.** As Table 5, but observed and simulated XCO (ppb) fields over the six selected areas: SEN, ESB, AMZ, SAS, CAF, and SBI between 2013 and 2015.

| Statistics | | WCA | ESB | AMZ | SAS | CAF | SBI |
|---|---|---|---|---|---|---|---|
| Mean (ppb) | | | | | | | |
| | MOPITT | 93.4 | 103.6 | 98.2 | 110.6 | 114.8 | 90.5 |
| | EsmAve | 95.2 | 103.7 | 101.1 | 120.4 | 115.1 | 99.4 |
| | GlcGlob | 97.4 | 105.9 | 104.9 | 124.7 | 123.0 | 102.8 |
| | GlcGeoc | 91.0 | 98.2 | 93.5 | 114.7 | 98.1 | 93.0 |
| | McdGlob | 99.1 | 109.1 | 107.6 | 125.5 | 130.9 | 104.7 |
| | McdGeoc | 92.9 | 101.2 | 97.3 | 116.6 | 102.7 | 96.4 |
| Standard deviation (ppb) | | | | | | | |
| | MOPITT | 11.5 | 10.4 | 23.5 | 23.4 | 28.7 | 29.5 |
| | EsmAve | 6.9 | 8.7 | 22.2 | 22.4 | 28.8 | 20.5 |
| | GlcGlob | 7.9 | 9.5 | 22.8 | 23.8 | 32.8 | 24.2 |
| | GlcGeoc | 5.9 | 7.5 | 20.0 | 20.9 | 19.0 | 14.7 |
| | McdGlob | 7.8 | 10.2 | 23.6 | 23.7 | 37.5 | 25.0 |
| | McdGeoc | 6.2 | 8.2 | 22.5 | 21.3 | 21.7 | 17.1 |
| Mean absolute error (ppb) | | | | | | | |
| | EsmAve | 5.8 | 4.6 | 8.9 | 12.8 | 12.2 | 13.7 |
| | GlcGlob | 5.9 | 5.1 | 10.0 | 15.0 | 13.7 | 15.0 |
| | GlcGeoc | 6.5 | 6.6 | 9.9 | 11.1 | 18.6 | 12.7 |
| | McdGlob | 7.1 | 6.7 | 11.7 | 16.3 | 18.4 | 16.2 |
| | McdGeoc | 6.3 | 5.3 | 8.9 | 11.6 | 15.5 | 13.3 |
| Correlation coefficient | | | | | | | |
| | EsmAve | 0.83 | 0.80 | 0.85 | 0.86 | 0.85 | 0.91 |
| | GlcGlob | 0.86 | 0.80 | 0.85 | 0.88 | 0.85 | 0.93 |
| | GlcGeoc | 0.76 | 0.75 | 0.79 | 0.81 | 0.79 | 0.82 |
| | McdGlob | 0.85 | 0.80 | 0.86 | 0.87 | 0.86 | 0.94 |
| | McdGeoc | 0.76 | 0.76 | 0.83 | 0.82 | 0.83 | 0.87 |
| Modified index of agreement | | | | | | | |
| | EsmAve | 0.62 | 0.70 | 0.74 | 0.65 | 0.73 | 0.59 |
| | GlcGlob | 0.64 | 0.68 | 0.71 | 0.62 | 0.71 | 0.58 |
| | GlcGeoc | 0.54 | 0.58 | 0.72 | 0.68 | 0.59 | 0.58 |
| | McdGlob | 0.58 | 0.60 | 0.68 | 0.59 | 0.65 | 0.56 |
| | McdGeoc | 0.55 | 0.64 | 0.75 | 0.67 | 0.65 | 0.58 |


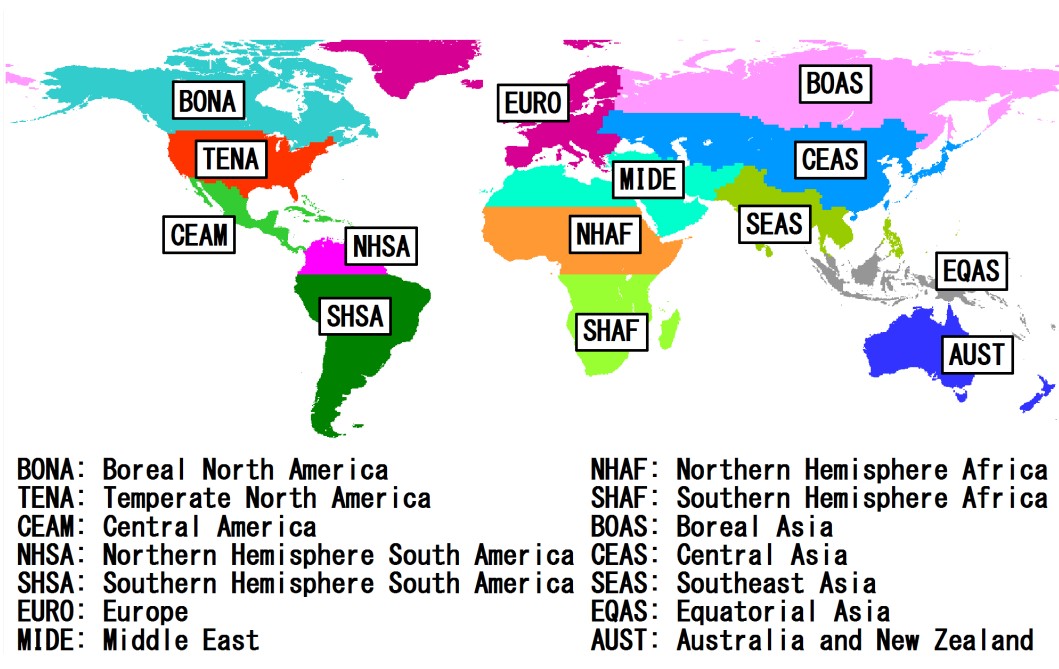

**Figure A1.** Map of the 14 global regions derived from Giglio et al. (2006) and van der Werf et al. (2017).

**Table A1.** BE and EF of CO (g CO kg$^{-1}$) for the LCC types used in GLC2000. See Fig. A1 for abbreviations of the 14 global regions. Letters in brackets show corresponding biome types from van der Werf et al. (2017); A: Boreal forest; B: Temperate forest; C: Tropical forest; D: Savanna; E: Peat; and F: Agriculture.

| LCC | BE | EF in the 14 regions | | | | | | | | | | | | | |
|---|---|---|---|---|---|---|---|---|---|---|---|---|---|---|---|
| | | BONE | TENA | CEAM | NHSA | SHSA | EURO | MIDE | NHAF | SHAF | BOAS | CEAS | SEAS | EQAS | AUST |
| Tree Cover, Broadleaved, Evergreen | 0.25 | 127 (A) | 88 (B) | 93 (C) | 93 (C) | 93 (C) | 88 (B) | 93 (C) | 93 (C) | 93 (C) | 127 (A) | 88 (B) | 93 (C) | 210 (E) | 88 (B) |
| Tree Cover, Broadleaved, Deciduous, Closed | 0.25 | 127 (A) | 88 (B) | 93 (C) | 93 (C) | 93 (C) | 88 (B) | 93 (C) | 93 (C) | 93 (C) | 127 (A) | 88 (B) | 93 (C) | 210 (E) | 88 (B) |
| Tree Cover, Broadleaved, Deciduous, Open | 0.4 | 127 (A) | 88 (B) | 93 (C) | 93 (C) | 93 (C) | 88 (B) | 93 (C) | 93 (C) | 93 (C) | 127 (A) | 88 (B) | 93 (C) | 210 (E) | 88 (B) |
| Tree Cover, Needle–Leaved, Evergreen | 0.25 | 127 (A) | 88 (B) | 93 (C) | 93 (C) | 93 (C) | 88 (B) | 93 (C) | 93 (C) | 93 (C) | 127 (A) | 88 (B) | 93 (C) | 210 (E) | 88 (B) |
| Tree Cover, Needle–Leaved, Deciduous | 0.25 | 127 (A) | 88 (B) | 93 (C) | 93 (C) | 93 (C) | 88 (B) | 93 (C) | 93 (C) | 93 (C) | 127 (A) | 88 (B) | 93 (C) | 210 (E) | 88 (B) |
| Tree Cover, Mixed Leaf Type | 0.25 | 127 (A) | 88 (B) | 93 (C) | 93 (C) | 93 (C) | 88 (B) | 93 (C) | 93 (C) | 93 (C) | 127 (A) | 88 (B) | 93 (C) | 210 (E) | 88 (B) |
| Tree Cover, Regularly Flooded, Fresh Water | 0 | 127 (A) | 88 (B) | 93 (C) | 93 (C) | 93 (C) | 88 (B) | 93 (C) | 93 (C) | 93 (C) | 127 (A) | 88 (B) | 93 (C) | 210 (E) | 88 (B) |
| Tree Cover, Regularly Flooded, Saline Water | 0 | 127 (A) | 88 (B) | 93 (C) | 93 (C) | 93 (C) | 88 (B) | 93 (C) | 93 (C) | 93 (C) | 127 (A) | 88 (B) | 93 (C) | 210 (E) | 88 (B) |
| Mosaic: Tree Cover, Other Natural Vegetation | 0.35 | 127 (A) | 88 (B) | 93 (C) | 93 (C) | 93 (C) | 88 (B) | 93 (C) | 93 (C) | 93 (C) | 127 (A) | 88 (B) | 93 (C) | 210 (E) | 88 (B) |
| Tree Cover, Burned | 0 | 127 (A) | 88 (B) | 93 (C) | 93 (C) | 93 (C) | 88 (B) | 93 (C) | 93 (C) | 93 (C) | 127 (A) | 88 (B) | 93 (C) | 210 (E) | 88 (B) |
| Shrub Cover, Closed–Open, Evergreen | 0.9 | 63 (D) | 63 (D) | 63 (D) | 63 (D) | 63 (D) | 63 (D) | 63 (D) | 63 (D) | 63 (D) | 63 (D) | 63 (D) | 63 (D) | 63 (D) | 63 (D) |
| Shrub Cover, Closed–Open, Deciduous | 0.4 | 63 (D) | 63 (D) | 63 (D) | 63 (D) | 63 (D) | 63 (D) | 63 (D) | 63 (D) | 63 (D) | 63 (D) | 63 (D) | 63 (D) | 63 (D) | 63 (D) |
| Herbaceous Cover, Closed–Open | 0.9 | 63 (D) | 63 (D) | 63 (D) | 63 (D) | 63 (D) | 63 (D) | 63 (D) | 63 (D) | 63 (D) | 63 (D) | 63 (D) | 63 (D) | 63 (D) | 63 (D) |
| Sparse Herbaceous or Sparse Shrub Cover | 0.6 | 63 (D) | 63 (D) | 63 (D) | 63 (D) | 63 (D) | 63 (D) | 63 (D) | 63 (D) | 63 (D) | 63 (D) | 63 (D) | 63 (D) | 63 (D) | 63 (D) |
| Regularly Flooded Shrub and/or Herbaceous Cover | 0 | 63 (D) | 63 (D) | 63 (D) | 63 (D) | 63 (D) | 63 (D) | 63 (D) | 63 (D) | 63 (D) | 63 (D) | 63 (D) | 63 (D) | 63 (D) | 63 (D) |
| Cultivated and Managed Areas | 0.6 | 102 (F) | 102 (F) | 102 (F) | 102 (F) | 102 (F) | 102 (F) | 102 (F) | 102 (F) | 102 (F) | 102 (F) | 102 (F) | 102 (F) | 102 (F) | 102 (F) |
| Mosaic: Cropland, Tree Cover, Other Natural Vegetation | 0.8 | 102 (F) | 102 (F) | 102 (F) | 102 (F) | 102 (F) | 102 (F) | 102 (F) | 102 (F) | 102 (F) | 102 (F) | 102 (F) | 102 (F) | 102 (F) | 102 (F) |
| Mosaic: Cropland, Shrub and/or Grass Cover | 0.75 | 102 (F) | 102 (F) | 102 (F) | 102 (F) | 102 (F) | 102 (F) | 102 (F) | 102 (F) | 102 (F) | 102 (F) | 102 (F) | 102 (F) | 102 (F) | 102 (F) |
| Bare Areas | 0 | 0 | 0 | 0 | 0 | 0 | 0 | 0 | 0 | 0 | 0 | 0 | 0 | 0 | 0 |
| Water Bodies | 0 | 0 | 0 | 0 | 0 | 0 | 0 | 0 | 0 | 0 | 0 | 0 | 0 | 0 | 0 |
| Snow and Ice | 0 | 0 | 0 | 0 | 0 | 0 | 0 | 0 | 0 | 0 | 0 | 0 | 0 | 0 | 0 |
| Artificial Surfaces and Associated Areas | 0 | 0 | 0 | 0 | 0 | 0 | 0 | 0 | 0 | 0 | 0 | 0 | 0 | 0 | 0 |
| No Data | 0 | 0 | 0 | 0 | 0 | 0 | 0 | 0 | 0 | 0 | 0 | 0 | 0 | 0 | 0 |

**Table A2.** As Table A1, but for the LCC types used in MCD12Q1.

| LCC | BE | EF in 14 regions | | | | | | | | | | | | | |
|---|---|---|---|---|---|---|---|---|---|---|---|---|---|---|---|
| | | BONE | TENA | CEAM | NHSA | SHSA | EURO | MIDE | NHAF | SHAF | BOAS | CEAS | SEAS | EQAS | AUST |
| Evergreen Needleleaf Forests | 0.25 | 127 (A) | 88 (B) | 93 (C) | 93 (C) | 93 (C) | 88 (B) | 93 (C) | 93 (C) | 93 (C) | 127 (A) | 88 (B) | 93 (C) | 210 (E) | 88 (B) |
| Evergreen Broadleaf Forests | 0.25 | 127 (A) | 88 (B) | 93 (C) | 93 (C) | 93 (C) | 88 (B) | 93 (C) | 93 (C) | 93 (C) | 127 (A) | 88 (B) | 93 (C) | 210 (E) | 88 (B) |
| Deciduous Needleleaf Forests | 0.25 | 127 (A) | 88 (B) | 93 (C) | 93 (C) | 93 (C) | 88 (B) | 93 (C) | 93 (C) | 93 (C) | 127 (A) | 88 (B) | 93 (C) | 210 (E) | 88 (B) |
| Deciduous Broadleaf Forests | 0.25 | 127 (A) | 88 (B) | 93 (C) | 93 (C) | 93 (C) | 88 (B) | 93 (C) | 93 (C) | 93 (C) | 127 (A) | 88 (B) | 93 (C) | 210 (E) | 88 (B) |
| Mixed Forests | 0.25 | 127 (A) | 88 (B) | 93 (C) | 93 (C) | 93 (C) | 88 (B) | 93 (C) | 93 (C) | 93 (C) | 127 (A) | 88 (B) | 93 (C) | 210 (E) | 88 (B) |
| Closed Shrublands | 0.9 | 63 (D) | 63 (D) | 63 (D) | 63 (D) | 63 (D) | 63 (D) | 63 (D) | 63 (D) | 63 (D) | 63 (D) | 63 (D) | 63 (D) | 63 (D) | 63 (D) |
| Open Shrublands | 0.9 | 63 (D) | 63 (D) | 63 (D) | 63 (D) | 63 (D) | 63 (D) | 63 (D) | 63 (D) | 63 (D) | 63 (D) | 63 (D) | 63 (D) | 63 (D) | 63 (D) |
| Woody Savannas | 0.8 | 63 (D) | 63 (D) | 63 (D) | 63 (D) | 63 (D) | 63 (D) | 63 (D) | 63 (D) | 63 (D) | 63 (D) | 63 (D) | 63 (D) | 63 (D) | 63 (D) |
| Savannas | 0.8 | 63 (D) | 63 (D) | 63 (D) | 63 (D) | 63 (D) | 63 (D) | 63 (D) | 63 (D) | 63 (D) | 63 (D) | 63 (D) | 63 (D) | 63 (D) | 63 (D) |
| Grasslands | 0.75 | 63 (D) | 63 (D) | 63 (D) | 63 (D) | 63 (D) | 63 (D) | 63 (D) | 63 (D) | 63 (D) | 63 (D) | 63 (D) | 63 (D) | 63 (D) | 63 (D) |
| Permanent Wetlands | 0 | 0 | 0 | 0 | 0 | 0 | 0 | 0 | 0 | 0 | 0 | 0 | 0 | 0 | 0 |
| Croplands | 0.8 | 102 (F) | 102 (F) | 102 (F) | 102 (F) | 102 (F) | 102 (F) | 102 (F) | 102 (F) | 102 (F) | 102 (F) | 102 (F) | 102 (F) | 102 (F) | 102 (F) |
| Urban and Built-Up Land | 0 | 0 | 0 | 0 | 0 | 0 | 0 | 0 | 0 | 0 | 0 | 0 | 0 | 0 | 0 |
| Cropland/Natural Vegetation Mosaics | 0.8 | 102 (F) | 102 (F) | 102 (F) | 102 (F) | 102 (F) | 102 (F) | 102 (F) | 102 (F) | 102 (F) | 102 (F) | 102 (F) | 102 (F) | 102 (F) | 102 (F) |
| Permanent Snow and Ice | 0 | 0 | 0 | 0 | 0 | 0 | 0 | 0 | 0 | 0 | 0 | 0 | 0 | 0 | 0 |
| Barren | 0.75 | 63 (D) | 63 (D) | 63 (D) | 63 (D) | 63 (D) | 63 (D) | 63 (D) | 63 (D) | 63 (D) | 63 (D) | 63 (D) | 63 (D) | 63 (D) | 63 (D) |
| Water Bodies | 0 | 0 | 0 | 0 | 0 | 0 | 0 | 0 | 0 | 0 | 0 | 0 | 0 | 0 | 0 |