# Peer review of "Sensitivity of biomass burning emissions estimates to land surface information"

_Biogeosciences, 2021_

## Referee Comment (RC1)

The authors calculate four sets of global biomass burning (BB) emission estimates with bottom-up approach, through combining two different aboveground biomass (AGB) with two different land cover classification (LCC) datasets. Modeled atmospheric CO concentrations from four simulations that use these four sets of BB emission estimates are then compared with observations. This study is a useful reference for the large uncertainties introduced to BB emission estimates (and hence model simulations) from AGB and LCC. The following comments need to be addressed.

1. I found the abstract a little hard to understand without reading through the main text. Please consider revising it.
   - For example, "We estimated four types of global BB emissions" is misleading and I suggest change it to "We developed four sets of global biomass burning emission estimates".
   - "We found a substantial spatial difference in CO emissions for both the AGB and LCC data …" This sentence can be changed to "We found a substantial spatial difference in CO emissions result from the use of the AGB and/or LCC data".
   - "At ground-based observation sites during fire seasons, statistical comparisons indicated that the impact of differences in the BB emissions estimates on atmospheric CO variability was poorly defined in our simulations." This sentence is very hard to understand without reading through the main text.
   - Consider adding some numbers in the abstract. For example "The mean annual BB emissions with their standard deviation are 526±53, 219±35, 624±57, and 293±44 Tg CO yr−1, for GlcGlob, GlcGeoc, McdGlob, and McdGeoc, respectively."

2. In the introduction, please acknowledge previous studies that attempted to address or quantify the uncertainties in BB emission estimates caused by land surface information (including but not limited to AGB, LCC, burned area etc) if there's any.

3. Line 81: "If a 1-km resolution grid point showed a fire flag on a particular day, then the 4 surrounding sub-grids with a 500 m resolution, located within the original MOD14A1 grid, were assumed to have burned; that is, $BA_{i,k}$ = 250,000 m2." I'm a little confused here. Shouldn't it be $BA_{i,k}$ =(500+500)x(500+500) m2=1e6 m2?

4. Section 2.3: I'm wondering why the authors only pick three sites. And what are the reason(s) for choosing these three specific sites?

5. Line 125: Need more information on the description of MOPITT. For example, which MOPITT version are you using. And "mean XCO distribution products (L3V95.6.3; Deeter et al., 2014) retrieved from both thermal infrared and near infrared observations", do you mean you used both TIR and NIR products separately, or used the joint product (JIR)? The model results need to be smoothed with MOPITT averaging kernel and prior profile before comparing to MOPITT, if you have done so please state it here for clarification.

6. Results: Please consider separate the results section to two sub-sections "comparisons of emission estimates" and "comparisons of model results".

7. Line 145: Figure 1 is for the year of 2009. While this info is included in the figure caption, it needs to be also included here in the main text.

8. Adding figures of global maps for the four datasets you use would be more intuitive for readers (2 LCC and 2 AGB).

9. Figure 2: This figure provides mean and standard deviation of the four BB emissions estimates. While this is useful, please also provide maps for the 4 estimates so that it's more intuitive and straightforward.

10. The authors compared their global total CO emission estimates with that from Andreae (2019), which is very useful. It would be nice if the authors can also provide comparisons of regional total CO emission estimates to other emission inventories. The main goal of this paper is to demonstrate the large differences in the emission estimates caused by AGB and LCC. For this purpose, Figures 2 & 3 are not enough. It would be helpful to show more detailed comparisons of the four estimates. For example, results for more regions such as those in which the three ground sites locate, or the six regions highlighted in Figure 6, or the 14 GFED regions.

11. Line 190: Change "we compared each BB estimate in the atmospheric CO field" to "we compared modeled CO fields using each BB emission estimate."

12. Section 3: If I understand correctly, the four emission estimates are monthly. Comparing model results driven by monthly fire emissions to daily observations is not informative as fire events are episodic. Model will not capture fire-induced daily variability from observational data. As shown in Figure 5, the model-observation agreement is poor. The author also noted this issue in the Discussion section. I suggest either (1) downscale the emission estimates to daily resolution with active fire detection satellite products and re-run the model, or (2) average the daily observations to monthly before doing the comparisons.

13. Line 221: Please change "Monthly mean XCO variations and the root mean square error (RMSE, ppb)" to "Monthly mean XCO and the root mean square error (RMSE, ppb)". It was confusing.

14. Line 222: I'm wondering if there is a specific reason for choosing southeast North America instead of Pacific Northwest?

---

## Author Comment (AC1)

Saito et al., 8 August 2021

MS number: bg-2021-130

Authors: M. Saito, T. Shiraishi, R. Hirata, Y. Niwa, K. Saito, M. Steinbacher, D. Worth, and T. Matsunaga

Title: Sensitivity of biomass burning emissions estimates to land surface information

Authors' response to anonymous referee #2

We appreciate the quick review from the anonymous referee #2. The comments are helpful to improve our manuscript, and we will revise the manuscript by following the referee's comments.

In the manuscript entitled "Sensitivity of biomass burning emissions estimates to land surface information", authors explored the sensitivity of biomass burning emission estimates to land surface information under four scenarios, based on land use and aboveground biomass. This paper just provided the results of different scenarios. It seems not a sensitivity analysis. The quantification results of sensitivity indicators are not outstanding. The sensitivity of quantified single parameter could be considered.

We approve that quantitative analyses regarding the sensitivity of individual variables could help to emphasize our results and improve the manuscript. We will additionally analyze this point and describe the results in the revised manuscript.

1. How to explain the meaning of equation 2?

   Eq. (2) shows variability of flammable fuels as a function of burning efficiency, above-ground biomass, and number of fire occurrence. We will add short explanation on Eq. (2).

2. Which variable of the formula can LCC and ACB provide data for? Please explain the relationship between LCC/ACB and emission estimation in detail.

   We agree with the comment that the relationship between LCC/AGB and emission estimates is not clear in the manuscript. We will clarify it.

3. What is the difference between providing data by MCD12Q1 and MOD14A1?

   Both MCD12Q1 and MOD14A1 are MODIS products, but the former is a product to provide global land cover types at yearly intervals and the latter is one for thermal anomalies and fire events. In our study, MCD12Q1 is used for land cover classification, whereas MOD14A1 is used to estimate changes in burned area.

4. Please supplement the discussion of the proportion of various types of biomass emissions (e.g., crop, forest, etc.) under different scenario for different LCC and AGB, and compare them with other studies.

We agree with the reviewer on comparing different emission scenarios based on land cover types. We will add this analysis in the revised manuscript.

5. In this paper, the different types of crops straws were not considered in the emission factors and activity data. There is comparable difference in emission factors of various crops straw, such as corn, rice, wheat, etc.

Emission factor could differ among different types of crops, while fixed emission factors are applied in our study. This is because that it is not able to obtain information on types of crops from the LCC data we used. As the reviewer suggested, this 'rough' LCC classification could be a factor of uncertainty in emission estimates, and we will discuss this point on the impact of different types of crops to biomass burning emissions by citing corresponding previous studies.

6. In this paper, MNM is set as the background station, whether it is similar to other pollution sources of the other two stations.

The MNM station is located in an isolated island over the Pacific and observation data at this station can be generally treated as background variability. We will add additional explanation on the MNM station.

7. Line 105-106, why the concentration of CH4 is fixed and what is the basis for setting it?

There are no observation data covering global distribution of atmospheric $CH_4$ variability including its vertical profile information, and the variability in the upper troposphere and stratosphere is little known. Thus, we decided to use fixed $CH_4$ concentration in our model simulation. The value of 1,800 ppb is maximum concentration of 'whole-atmosphere monthly mean $CH_4$ concentration' derived from Greenhouse gases Observing SATellite (GOSAT; Yokota et al., 2009) observations in the period 2009 - 2015. We agree that the manuscript should be revised to explain this point.

Yokota, T., et al. "Global concentrations of CO2 and CH4 retrieved from GOSAT: First preliminary results." SOLA 5 (2009): 160-163.

8. Line 194-195, the data is not well expressed.

This suggestion is not clear, but we will describe additional explanation for Table 4.

9. Fig.4: Incomplete display of broken line chart.

We could not correctly comprehend this comment, because broken lines are not used in Fig. 4.

---

## Author Comment (AC2)

Saito et al., 8 August 2021

MS number: bg-2021-130

Authors: M. Saito, T. Shiraishi, R. Hirata, Y. Niwa, K. Saito, M. Steinbacher, D. Worth, and T. Matsunaga

Title: Sensitivity of biomass burning emissions estimates to land surface information

Authors' response to anonymous referee #1

We truly appreciate all the constructive comments and suggestions from the reviewer. The comments are very helpful to improve our manuscript, and we will adopt all the suggestions in our revised manuscript.

The authors calculate four sets of global biomass burning (BB) emission estimates with bottomup approach, through combining two different aboveground biomass (AGB) with two different land cover classification (LCC) datasets. Modeled atmospheric CO concentrations from four simulations that use these four sets of BB emission estimates are then compared with observations. This study is a useful reference for the large uncertainties introduced to BB emission estimates (and hence model simulations) from AGB and LCC. The following comments need to be addressed.

1. I found the abstract a little hard to understand without reading through the main text. Please consider revising it.

   - For example, "We estimated four types of global BB emissions" is misleading and I suggest change it to "We developed four sets of global biomass burning emission estimates".

     We appreciate the suggestion; we corrected the text.

   - "We found a substantial spatial difference in CO emissions for both the AGB and LCC data ..." This sentence can be changed to "We found a substantial spatial difference in CO emissions result from the use of the AGB and/or LCC data".

     We appreciate the suggestion; we corrected the text.

   - "At ground-based observation sites during fire seasons, statistical comparisons indicated that the impact of differences in the BB emissions estimates on atmospheric CO variability was poorly defined in our simulations." This sentence is very hard to understand without reading through the main text.

     We appreciate the comment; we will revise the manuscript.

   - Consider adding some numbers in the abstract. For example "The mean annual BB emissions with their standard deviation are 526±53, 219±35,

624±57, and 293$pm$44 Tg CO yr$^1$, for GlcGlob, GlcGeoc, McdGlob, and McdGeoc, respectively."

We agree with the suggestion and will revise the abstract to describe the differences in emissions estimate using some numbers.

2. In the introduction, please acknowledge previous studies that attempted to address or quantify the uncertainties in BB emission estimates caused by land surface information (including but not limited to AGB, LCC, burned area etc) if there's any.

We agree with this point and will write acknowledgement to previous studies in the introduction.

3. Line 81: "If a 1-km resolution grid point showed a fire flag on a particular day, then the 4 surrounding sub-grids with a 500 m resolution, located within the original MOD14A1 grid, were assumed to have burned; that is, BAi,k = 250,000 m$^2$." I'm a little confused here. Shouldn't it be BAi,k = (500+500) × (500+500) m$^2$ = 1e6 m$^2$?

MOD14A1 provides fire mask data at a 1-km resolution, while our study estimates biomass burning emissions with a spatial resolution of 500 m. We regridded the MOD14A1 data to a 500-m resolution without interpolation, then the regridded fire mask data were used as $\text{BA}_{i,k}$. We approve that the sentence in Line 81 confuses the readers and will revise it.

4. Section 2.3: I'm wondering why the authors only pick three sites. And what are the reason(s) for choosing these three specific sites?

The MNM site is located in an isolated island and we think that the site is an appropriate location for representing background variations of atmospheric CO concentrations. We investigated all available observation sites on the WDCGG website, then the BKT and ETL sites were chosen as representative observation sites clearly showing the influence of fire events in the variability of atmospheric CO concentrations. We will clarify this point in the revised manuscript.

5. Line 125: Need more information on the description of MOPITT. For example, which MOPITT version are you using. And "mean XCO distribution products (L3V95.6.3; Deeter et al., 2014) retrieved from both thermal infrared and near infrared observations", do you mean you used both TIR and NIR products separately, or used the joint product (JIR)? The model results need to be smoothed with MOPITT averaging kernel and prior profile before comparing to MOPITT, if you have done so please state it here for clarification.

We really appreciate this important comment. In this study, multispectral TIR/NIR retrievals were used, but MOPITT averaging kernels were not taken into account for model simulations by assuming that the total

column averaging kernels of MOPITT are near unity over the altitude. However, as mentioned by the reviewer, MOPITT user's guide shows sensitivity of the averaging kernels to the altitude, so that we will reconfigure the model simulation to include the MOPITT averaging kernels and the prior information.

6. Results: Please consider separate the results section to two sub-sections "comparisons of emission estimates" and "comparisons of model results".

We agree with the reviewer on separating the results section to two subsections.

7. Line 145: Figure 1 is for the year of 2009. While this info is included in the figure caption, it needs to be also included here in the main text.

We agree with the reviewer and will add an explanation on Fig. 1 in the revised manuscript.

8. Adding figures of global maps for the four datasets you use would be more intuitive for readers (2 LCC and 2 AGB).

We agree with this suggestion and will add the figures.

9. Figure 2: This figure provides mean and standard deviation of the four BB emissions estimates. While this is useful, please also provide maps for the 4 estimates so that it's more intuitive and straightforward.

Thanks for this suggestion; we will show maps for the 4 emissions estimates in the revised manuscript.

10. The authors compared their global total CO emission estimates with that from Andreae (2019), which is very useful. It would be nice if the authors can also provide comparisons of regional total CO emission estimates to other emission inventories. The main goal of this paper is to demonstrate the large differences in the emission estimates caused by AGB and LCC. For this purpose, Figures 2 & 3 are not enough. It would be helpful to show more detailed comparisons of the four estimates. For example, results for more regions such as those in which the three ground sites locate, or the six regions highlighted in Figure 6, or the 14 GFED regions.

We agree with that it is necessary to compare our emissions estimates with other emissions inventories over regional scale for clarifying the differences between our estimates and other inventories. We will add new analyses regarding this point and revise Figs. 2 and 3 in accordance with the reviewer's suggestion.

11. Line 190: Change "we compared each BB estimate in the atmospheric CO field" to "we compared modeled CO fields using each BB emission estimate."

We appreciate the suggestion; we corrected the text.

12. Section 3: If I understand correctly, the four emission estimates are monthly. Comparing model results driven by monthly fire emissions to daily observations is not informative as fire events are episodic. Model will not capture fire-induced daily variability from observational data. As shown in Figure 5, the model-observation agreement is poor. The author also noted this issue in the Discussion section. I suggest either (1) downscale the emission estimates to daily resolution with active fire detection satellite products and re-run the model, or (2) average the daily observations to monthly before doing the comparisons.

As suggested by the reviewer, the atmospheric tracer transport model NICAM-TM was operated with a time resolution of 20 minutes, whereas input BB emission estimates have a monthly resolution. We admit that the monthly resolution of BB emissions estimates is insufficient to simulate daily variations of atmospheric CO concentrations at ground-based observation sites. Considering the amount of work, we would like to attempt (2) in the revision.

13. Line 221: Please change "Monthly mean XCO variations and the root mean square error (RMSE, ppb)" to "Monthly mean XCO and the root mean square error (RMSE, ppb)". It was confusing.

We appreciate the suggestion; we corrected the text.

14. Line 222: I'm wondering if there is a specific reason for choosing southeast North America instead of Pacific Northwest?

We chose simply the southeast North America as a higher BB emissions area, but it is certainly true that the Pacific Northwest has been affected by sever wildfires. We will analyze the Pacific Northwest and revise the manuscript.

---

## Author Response (AR1)

Saito et al. 15 Nov. 2021

MS number: bg-2021-130

Authors: M. Saito, T. Shiraishi, R. Hirata, Y. Niwa, K. Saito, M. Steinbacher, D. Worthy, and T. Matsunaga

Title: Sensitivity of biomass burning emissions estimates to land surface information

Dear Editor and Associate Editor

We prepared a revised manuscript for submission bg-2021-130 'Sensitivity of biomass burning emissions estimates to land surface information'. Enclosed are the revised manuscript and the response letter to the reviewers. We appreciated the insightful comments of the reviewers, that truly helped us to improve the manuscript. We tried to follow the reviewer's comments and suggestions at best. The main revisions are listed in this letter, followed by a more detailed response in the appendices:

- The methods section was extended with a new subsection to describe how we incorporated MOPITT averaging kernels in our XCO simulations.

- As the MOPITT averaging kernels were not incorporated in our simulation results of previous manuscript, we re-simulated atmospheric CO variability and incorporated MOPITT averaging kernels in our simulated XCO. In this re-simulation process, a mistake in a model initial setting associated with model restart file; in previous simulation, simulated atmospheric field had been reset to initial setting at the beginning of every year, was found. In this revision, we present the results based on a fixed model setting that simulates continuously atmospheric variability for the period 2009–2015.

- In response to the reviewers, comparison of our biomass burning emissions estimates with two reference inventories were newly added.

- All figures and Tables 4 and 5 (now Tables 5 and 6) were modified based on the re-simulation results, and Table 4 was newly added to compare our results with reference inventories.

- Aboveground biomass and land cover classification maps used in this study were added as supplementary materials.

Thank you for your time and efforts and we hope to hear from you soon.

Kind regards,

Makoto Saito on behalf of the author team

**Reviewer #1**

1. I found the abstract a little hard to understand without reading through the main text. Please consider revising it.

   - For example, "We estimated four types of global BB emissions" is misleading and I suggest change it to "We developed four sets of global biomass burning emission estimates".

     We appreciate the suggestion; we corrected the text (line 2).

   - "We found a substantial spatial difference in CO emissions for both the AGB and LCC data ..." This sentence can be changed to "We found a substantial spatial difference in CO emissions result from the use of the AGB and/or LCC data".

     We appreciate the suggestion; we corrected the text as "Using the AGB and/or LCC data led to substantially different spatial estimates of CO emissions" (line 5–6).

   - "At ground-based observation sites during fire seasons, statistical comparisons indicated that the impact of differences in the BB emissions estimates on atmospheric CO variability was poorly defined in our simulations." This sentence is very hard to understand without reading through the main text.

     We appreciate the comment; we revised the sentence as "At ground-based observation sites during fire seasons, the impact of intermittent fire events was poorly defined in our simulations due to the coarse resolution, which obscured temporal and spatial variability in the simulated atmospheric CO concentration." (line 10–12).

   - Consider adding some numbers in the abstract. For example "The mean annual BB emissions with their standard deviation are 526±53, 219±35, 624±57, and 293$pm$44 Tg CO yr$^1$, for GlcGlob, GlcGeoc, McdGlob, and McdGeoc, respectively."

     We agree with the suggestion and added a sentence in abstract as "Using the AGB and/or LCC data led to substantially different spatial estimates of CO emissions, with a large (factor of approximately three) spread of estimates for the mean annual CO emissions; $526 \pm 53$, $219 \pm 35$, $624 \pm 57$, and $293 \pm 44$ Tg CO yr$^{-1}$, for GlcGlob, GlcGeoc, McdGlob, and McdGeoc, respectively, and $415 \pm 47$ Tg CO yr$^{-1}$ for their ensemble average (EsmAve)." (line 5–8).

2. In the introduction, please acknowledge previous studies that attempted to address or quantify the uncertainties in BB emission estimates caused by land surface information (including but not limited to AGB, LCC, burned area etc) if there's any.

   We agree with this point and added following sentences "These inventories provide BB emissions covering a variety of temporal and spatial resolutions and have facilitated improvements in our understanding of greenhouse gas emissions from BB. Research efforts, including the continuous

improvement of land surface information, are important to the study of the global carbon cycle, as they have reduced estimation uncertainties." (line 33–36).

3. Line 81: "If a 1-km resolution grid point showed a fire flag on a particular day, then the 4 surrounding sub-grids with a 500 m resolution, located within the original MOD14A1 grid, were assumed to have burned; that is, BAi,k = 250,000 m$^2$." I'm a little confused here. Shouldn't it be BAi,k = (500+500) × (500+500) m$^2$ = 1e6 m$^2$?

MOD14A1 provides fire mask data at a 1-km resolution, while our study estimates biomass burning emissions with a spatial resolution of 500 m. We re-gridded the MOD14A1 data to a 500-m resolution without interpolation, then the re-gridded fire mask data were used as $BA_{i,k}$. We approve that the sentence in Line 81 confuses the readers and revised it as "This study disaggregated the MOD14A1 product into a 500 m resolution, and assigned $BA_{i,k} = 250,000$ m$^2$ for any grid showing a fire flag." (line 86–87).

4. Section 2.3: I'm wondering why the authors only pick three sites. And what are the reason(s) for choosing these three specific sites?

We revised the manuscript as "From the 28 WDCGG stations with hourly observation data, we selected those showing abrupt increases in atmospheric CO concentrations during dry seasons. Consequently, two ground-based sites: Bukit Kototabang, Indonesia (BKT, 0.20°S, 100.32°E; Zellweger et al., 2019) and East Trout Lake, Canada (ETL, 54.35°N, 104.99°W; Kim, 2016) were selected to evaluate the estimates of BB emissions from local fire events. Minamitorishima, Japan (MNM, 24.29°N, 153.98°E; Watanabe et al., 2000) was also used as spatially representative background information." (line 126–130).

5. Line 125: Need more information on the description of MOPITT. For example, which MOPITT version are you using. And "mean XCO distribution products (L3V95.6.3; Deeter et al., 2014) retrieved from both thermal infrared and near infrared observations", do you mean you used both TIR and NIR products separately, or used the joint product (JIR)? The model results need to be smoothed with MOPITT averaging kernel and prior profile before comparing to MOPITT, if you have done so please state it here for clarification.

We incorporated MOPITT averaging kernels and prior information in this revision. Subsection 2.4 was newly added to describe how to incorporate MOPITT averaging kernels and prior information into our simulations, and all of results, figures (Figs 6 and 7), and Table 5 (new Table 6) associated with this revision were revised.

6. Results: Please consider separate the results section to two sub-sections "comparisons of emission estimates" and "comparisons of model results".

We agree with the reviewer and separated the results section to two subsections.

7. Line 145: Figure 1 is for the year of 2009. While this info is included in the figure caption, it needs to be also included here in the main text.

We agree with the reviewer and added an explanation on Fig. 1 in the revised manuscript (line 188-190).

8. Adding figures of global maps for the four datasets you use would be more intuitive for readers (2 LCC and 2 AGB).

We agree with this suggestion and prepared supplemental materials to show the figures.

9. Figure 2: This figure provides mean and standard deviation of the four BB emissions estimates. While this is useful, please also provide maps for the 4 estimates so that it's more intuitive and straightforward.

Thanks for this suggestion; we revised Figure 2 according to this advice.

10. The authors compared their global total CO emission estimates with that from Andreae (2019), which is very useful. It would be nice if the authors can also provide comparisons of regional total CO emission estimates to other emission inventories. The main goal of this paper is to demonstrate the large differences in the emission estimates caused by AGB and LCC. For this purpose, Figures 2 & 3 are not enough. It would be helpful to show more detailed comparisons of the four estimates. For example, results for more regions such as those in which the three ground sites locate, or the six regions highlighted in Figure 6, or the 14 GFED regions.

We agree with that it is necessary to compare our emissions estimates with other emissions inventories over regional scale for clarifying the differences between our estimates and other inventories. We added new analyses regarding this point in line 251–274 and new Table 4.

11. Line 190: Change "we compared each BB estimate in the atmospheric CO field" to "we compared modeled CO fields using each BB emission estimate."

We appreciate the suggestion; we corrected the text (line 276).

12. Section 3: If I understand correctly, the four emission estimates are monthly. Comparing model results driven by monthly fire emissions to daily observations is not informative as fire events are episodic. Model will not capture fire-induced daily variability from observational data. As shown in Figure 5, the model-observation agreement is poor. The author also noted this issue in the Discussion section. I suggest either (1) downscale the emission estimates

to daily resolution with active fire detection satellite products and re-run the model, or (2) average the daily observations to monthly before doing the comparisons.

As suggested by the reviewer, the atmospheric tracer transport model NICAM-TM was operated with a time resolution of 20 minutes, whereas input BB emission estimates have a monthly resolution. We admit that the monthly resolution of BB emissions estimates is insufficient to simulate daily variations of atmospheric CO concentrations at ground-based observation sites. We remade Figure 5 using monthly averaged variations and revised corresponding sentences (line 291–300).

13. Line 221: Please change "Monthly mean XCO variations and the root mean square error (RMSE, ppb)" to "Monthly mean XCO and the root mean square error (RMSE, ppb)". It was confusing.

    We appreciate the suggestion; we corrected the text (line 315).

14. Line 222: I'm wondering if there is a specific reason for choosing southeast North America instead of Pacific Northwest?

    We chose simply the southeast North America as a higher BB emissions area, but it is certainly true that the Pacific Northwest has been affected by sever wildfires. We analyzed the Pacific Northwest in the revision (Figs 6a and 7a).

**Reviewer #2**

1. How to explain the meaning of equation 2?

   Eq. (2) shows variability of flammable fuels as a function of burning efficiency, above-ground biomass, and number of fire occurrence. We newly added short explanation on Eq. (2) in line 60–63.

2. Which variable of the formula can LCC and ACB provide data for? Please explain the relationship between LCC/ACB and emission estimation in detail.

   We agree with the comment that the relationship between LCC/AGB and emission estimates is not clear in the manuscript. We revised subsection 2.1 to clarify it.

3. What is the difference between providing data by MCD12Q1 and MOD14A1?

   Both MCD12Q1 and MOD14A1 are MODIS products, but the former is a product to provide global land cover types at yearly intervals and the latter is one for thermal anomalies and fire events. In our study, MCD12Q1 was used for land cover classification, whereas MOD14A1 was used to estimate changes in burned area.

4. Please supplement the discussion of the proportion of various types of biomass emissions (e.g., crop, forest, etc.) under different scenario for different LCC and AGB, and compare them with other studies.

We agree with the reviewer on comparing different emission scenarios based on land cover types. To balance with suggestion by reviewer 1, we added new analysis for 14 regions and comparison with reference inventories (line 251–274) in the revised manuscript.

5. In this paper, the different types of crops straws were not considered in the emission factors and activity data. There is comparable difference in emission factors of various crops straw, such as corn, rice, wheat, etc.

Emission factor could differ among different types of crops, while fixed emission factors were applied in our study. This is because that it is not able to obtain information on types of crops from the LCC data we used. As the reviewer suggested, this 'rough' LCC classification could be a factor of uncertainty in emission estimates, and we discussed this point on the impact of different types of crops to biomass burning emissions by citing corresponding previous studies (line 388–393).

6. In this paper, MNM is set as the background station, whether it is similar to other pollution sources of the other two stations.

The MNM station is located in an isolated island over the Pacific and observation data at this station can be generally treated as background variability. We added additional explanation on the MNM station (line 130–133).

7. Line 105-106, why the concentration of CH4 is fixed and what is the basis for setting it?

There are no observation data covering global distribution of atmospheric $CH_4$ variability including its vertical profile information, and the variability in the upper troposphere and stratosphere is little known. Thus, we decided to use fixed $CH_4$ concentration in our model simulation. The value of 1,800 ppb is maximum concentration of 'whole-atmosphere monthly mean $CH_4$ concentration' derived from Greenhouse gases Observing SATellite (GOSAT; Yokota et al., 2009) observations in the period 2009 - 2015. We revised the manuscript to explain this point (line 111-113).

Yokota, T., et al. "Global concentrations of CO2 and CH4 retrieved from GOSAT: First preliminary results." SOLA 5 (2009): 160-163.

8. Line 194-195, the data is not well expressed.

This suggestion is not clear, but we revised the manuscript to describe additional explanation for Table 4 (line 276–290).

9. Fig.4: Incomplete display of broken line chart.

   We could not correctly comprehend this comment, because broken lines are not used in Fig. 4. In this revision, all figures were remade based on the results of re-simulation, which will satisfy this reviewers's comment.

---

## Author Response (AR2)

Saito et al. 10 Jan. 2022

MS number: bg-2021-130

Authors: M. Saito, T. Shiraishi, R. Hirata, Y. Niwa, K. Saito, M. Steinbacher, D. Worthy, and T. Matsunaga

Title: Sensitivity of biomass burning emissions estimates to land surface information

Reviewers' Comments

**Report #1**

1. Fig.4: The value of black line has exceeded the upper limit of the ordinate, such as (a)BKT, the value from August to December.

   **In response to this comment, we extended y-axis ranges in Fig. 4a and b to show all variations of atmospheric CO concentrations.**

**Report #2**

1. Based on the info you provided (MOP03JM-L3V95.6.3), you are using the MOPITT Version 9 Level 3 multispectral (TIR/NIR) product. Please include this info. And the reference for MOPITT version 9 is not published yet. So please add this reference that describes some of the new changes in MOPITT version 9:
   Deeter MN, Mao D, Martínez-Alonso S, Worden HM, Andreae MO, Schlager H. Impacts of MOPITT cloud detection revisions on observation frequency and mapping of highly polluted scenes. Remote Sensing of Environment. 2021 Sep 1;262:112516.

   **We edited the explanation of MOPITT product we used and added new reference Deeter et al. (2021) (Lines 134–138/463–464 in the revised manuscript).**

2. Line 220: please change "average annual BB CO emissions" to "annual average of BB CO emissions"

   **We edited the sentence according the advise (Line 211 and caption of Fig. 2 in the revised manuscript).**

3. Line 245: figure 2b, do you mean figure 2f? Besides, the red box in figure 2f is hard to see without zooming in. Please try if changing the red box to black box helps.

   **Yes, correct figure number is Figure 2f. We corrected this mistake and changed color of the rectangle from red to black (Line 233, caption of Fig. 3, and Fig. 2f in the revised manuscript).**